

# 1   Leaf habit and nutrient availability drive leaf nutrient resorption globally

Gabriela Sophia[1,2,3], Silvia Caldararu[4], Benjamin D. Stocker[3,5], Sönke Zaehle[1,6]
[1] Max Planck for Biogeochemistry, Jena, Germany; [2] International Max Planck Research
School on Global Biogeochemical Cycles; [3] Geographisches Institut, Universität Bern,
Switzerland; [4] Discipline of Botany, School of Natural Sciences, Trinity College Dublin,
Dublin, Ireland; [5] Oeschger Center, Universität Bern, Switzerland; [6] Friedrich Schiller
Universität Jena, Jena, Germany; (gsophia@bgc-jena.mpg.de)

## 9   Abstract

Nutrient resorption from senescing leaves can significantly affect ecosystem nutrient cycling,
making it an essential process to better understand long-term plant productivity under
environmental change that affects the balance between nutrient availability and demand.
Although it is known that nutrient resorption rates vary strongly between different species
and across environmental gradients, the underlying driving factors are insufficiently
quantified. Here, we present an analysis of globally distributed observations of leaf nutrient
resorption to investigate the factors driving resorption efficiencies for nitrogen (NRE) and
phosphorus (PRE). Our results show that leaf structure and habit, together with indicators of
nutrient availability, are the two most important factors driving spatial variation in NRE.
Overall, we found higher NRE in deciduous plants (65.2% ± 12.4%, n=400) than in
evergreen plants (57.9% ± 11.4%, n=551), likely associated with a higher share of metabolic
N in leaves of deciduous plants. Tropical regions show the lowest resorption for N (NRE:
52.4% ± 12.1%) and tundra ecosystems in polar regions show the highest (NRE: 69.6% ±
12.8%), while the minimum PRE is in temperate regions (57.8% ± 13.6%) increasing to
boreal regions (67.3% ± 13.6%). Soil clay content, N and P atmospheric deposition - a
globally available proxy for soil fertility - and MAP played an important role in this pattern,
where we found higher NRE and PRE in high latitudes. The statistical relationships
developed in this analysis indicate an important role of leaf habit and type for nutrient cycling
and guide improved representations of plant-internal nutrient re-cycling and nutrient
conservation strategies in vegetation models.
**Keywords:** Leaf nutrient content; Leaf structure; Nitrogen and phosphorus resorption
efficiency; Plant ecophysiology; Plant functional traits; Plant nutrient limitation.





## 1. Introduction

Nutrient cycling plays an important role in shaping the global distribution of terrestrial primary productivity (Le Bauer et al., 2008; Zaehle, 2013; Du et al., 2020). Nitrogen (N) and phosphorus (P) are the main limiting nutrients for plant growth. N is needed to maintain and produce essential proteins for the biosynthesis; while P is an element of genetic material and plays a major role in the regeneration of the main receptor of carbon (C) assimilation, and in the production of energy that conducts many processes in living cells (Chapin, 1980; Güsewell, 2004). The anthropogenic increase in atmospheric $CO_2$ since the beginning of industrialization has the potential to enhance the terrestrial carbon sink through increasing plant photosynthetic rates, a process known as $CO_2$ fertilization (Bazzaz, 1990). A potential limitation to the fertilization effect is progressive nutrient limitation to growth (Luo et al., 2004) and associated plant strategies to deal with such limitations. Thus, understanding the ways in which nutrients circulate in ecosystems and are acquired, lost, and conserved by plants, is essential for simulating plant response to global changes.

Nutrient resorption - defined here as the translocation of nutrients from senescing leaves to temporary storage tissues - is a plant strategy for nutrient conservation (Killingbeck, 1996; Kobe et al., 2005). It allows plants to directly reuse nutrients, decreasing the dependence on soil nutrient availability and the competition for these nutrients with other plants and microbes, especially in nutrient-limited environments (Aerts, 1996; Aerts and Chapin, 1999). The question that arises is then why do plants not all resorb the entirety of leaf nutrients for being more efficient? The fact that they don't implies the existence of costs and limitations to resorption. A quantitative understanding of nutrient resorption can yield insights into plant strategies to cope with nutrient limitation (Aerts and Chapin, 1999; Chapin et al., 2011). This is because the resorption process influences most other ecosystem processes that determine plant growth, as it directly affects litter quality and therefore soil organic matter decomposition and has indirect consequences for plant nutrient uptake, carbon cycling and finally plant competition (Killingbeck, 1996; Berg and McClaugherty, 2008). The average fraction of leaf nutrients resorbed before abscission is estimated to be ~62% for N and ~65% for P (Vergutz et al., 2013). Cleveland et al. (2013) estimated that this corresponds to 31% of a plant's annual demand for N and 40% of the annual demand for P, but with large geographical and species variations.





However, despite advances in recent years, the drivers behind nutrient resorption and its
variation are still unclear: First, soil fertility has long been assumed to be a key driver for
variations in nutrient resorption, with increased resorption in infertile soils as the plant's main
strategy for nutrient conservation (Aerts and Chapin, 1999). Nonetheless, there is diverging
evidence established at different geographic scales, showing positive correlations (Aerts and
Chapin, 1999), negative correlations (Yuan and Chen, 2015; Xu et al., 2021), and even a lack
of correlation between soil fertility and resorption efficiency (Vergutz et al., 2013). Second,
climate factors are also considered to be important drivers for resorption, but the evidence is
equally conflicting: On the one hand, Yuan and Chen (2009) and Yan et al. (2017) suggested
NRE is decreasing with mean annual temperature (MAT) and precipitation (MAP), with the
opposite trend for PRE, arguing that colder regions tend to be more N-limited, while
P-limitation is observed more commonly in warmer environments. From low to high latitudes
globally, the role of N in limiting productivity tends to increase as the availability of N is
mainly determined by temperature-limited processes such as biological N fixation and
mineralization of soil organic matter (Cleveland et al., 2013; Fay et al., 2015; Deng et al.,
2018), but the presence of N fixers in tropical forests introduces complexity to the pattern of
nutrient limitation between tropical and temperate zones (Hedin et al., 2009). Nevertheless,
the limited availability of P in the tropics due to highly weathered soils distinguishes low- to
mid-latitude environments (Elser et al., 2007). On the other hand, Vergutz et al. (2013) and
Xu et al., 2021 showed that NRE and PRE are both increasing with decreasing MAT and
MAP toward higher latitudes.
A third set of studies suggests plant functional types, leaf stoichiometry and plant nutrient
demand as drivers for nutrient resorption (Reed et al., 2012; Han et al., 2013; Tang et al.,
2013; Brant and Chen, 2015; Du et al., 2020; Chen et al., 2021a; Sun et al., 2023). When
found greater nutrient resorption in evergreen species, it is assumed to be a conservation
strategy given their comparatively low leaf nutrient content and slow growth rate and
predominant occurrence in nutrient-limited biomes (Killingbeck, 1996; Yan et al., 2017; Xu
et al., 2021). The same argument has been used for interpreting differences between
broad-leaves and needle-leaves, in which nutrient resorption is generally observed to be
higher in needles as a strategy to acclimatize and survive in resource-limited environments
(Aerts and Chapin, 1999; Yuan et al., 2005; Yan et al., 2017; Xu et al., 2021). Previous
studies have suggested that shrub species generally display higher nutrient resorption rates



compared to trees, due to their smaller leaves with shorter life cycles and for the need to
optimize nutrient use in resource-limited environments (Killingbeck, 1996; Yuan and Chen,
2009; Yan et al., 2017; Xu et al., 2021). However, Brant and Chen (2015) suggest that
deciduous plants are more dependent on nutrient resorption as their investment in green leaf
nutrients is higher to maintain their fast growth through high physiological activity during the
growing season. Plants with a slow growth strategy, such as evergreens and needle-leaves,
have lower photosynthetic nutrient use efficiency due to a higher allocation of C and N to leaf
structural rather than metabolic compounds (Reich et al., 2017). Onoda et al. (2017)
empirically supports this by showing that a greater allocation of nutrients to structural
compounds is associated with decreased specific leaf area (SLA) and increased diffusive
limitation to photosynthesis. Thus, variations in leaf traits and construction costs could
contribute to differences in resorption between plant functional types (PFTs). Nevertheless,
Drenovsky et al. (2010; 2019) suggested that resorption variability is influenced by an
interplay of the discussed drivers, that includes soil properties, climatic conditions, and plant
characteristics. Estiarte et al. (2023) support that leaf biochemistry of plants determine the
first limitation to nutrient resorption, with a secondary regulation in resorption by
environmental conditions, while the costs of leaf aging remain consistent.
The divergence of observed patterns highlights the need for further investigation into the
main drivers of variations in nutrient resorption, distinguishing the influence of plant types,
soil and climatic conditions. In this study, we present a meta-analysis that combines the
version 5.0 of TRY Plant Trait database (Kattge et al., 2020) with different ancillary datasets
for climate and soil factors to investigate global patterns of resorption efficiencies for
nitrogen (NRE) and phosphorus (PRE). We aim to extend woody species observations for
nutrient resorption and investigate the factors that explain observed patterns along three main
axes: climate, soil fertility and leaf properties.

## 2. Methods

### 2.1 Data collection

We assembled the dataset from the TRY Plant Trait database (https://www.try-db.org, Kattge
et al., 2020, version 5.0) containing field measurements of paired leaf and litter mass-based
tissue N and P concentrations ($N_{\text{mass, leaf}}$, $P_{\text{mass, leaf}}$, $N_{\text{mass, litter}}$, $P_{\text{mass, litter}}$) to derive the fractional
nutrient resorption (described in Sect. 2.2), and plant functional traits recorded in parallel



from the same species and same location to consider as biological predictors variables (Table
1). As additional predictors for nutrient resorption, we combined it with climate and soil input
data (Table 2). We processed the data using R statistical software (version 4.0.4), keeping the
data at species-level. To manipulate the extracted functional traits, we used the package
{rtry} (Lam et al., 2022) developed to support the preprocessing of TRY Database (version
1.0.0), and {tidyverse} package (Wickham et al., 2019) with its dependencies (version 1.3.2).
The data processing followed the quality control according to the published protocol of TRY
(Kattge et al., 2011; 2020).

**Table 1.** Traits extracted from TRY database to derive nutrient resorption.

**Plant traits**

|  | Variable name | Unit |
|---|---|---|
| $N_{\text{mass, leaf}}$ | Leaf nitrogen (N) content per leaf dry mass | mg g |
| $P_{\text{mass, leaf}}$ | Leaf phosphorus (P) content per leaf dry mass | mg g |
| $N_{\text{mass, litter}}$ | Litter nitrogen (N) content per litter dry mass | mg g |
| $P_{\text{mass, litter}}$ | Litter phosphorus (P) content per litter dry mass | mg g |
| SLA | Leaf area per leaf dry mass: petiole, rhachis and midrib excluded | $mm^2\ mg^{-1}$ |
| SLA | Leaf area per leaf dry mass: petiole excluded | $mm^2\ mg^{-1}$ |
| SLA | Leaf area per leaf dry mass: petiole included | $mm^2\ mg^{-1}$ |
| SLA | Leaf area per leaf dry mass: undefined if petiole is in- or excluded | $mm^2\ mg^{-1}$ |
|  | Leaf dry mass | mg |
|  | Leaf senescent dry mass | mg |
| LML | Leaf Mass Loss | unitless |
| PFT | Plant functional type / growth form | unitless |
| KGC | Köppen Climate Classification | unitless |



As predictors, we used a set of climate variables, N and P deposition, vegetation type-related
variables, and soil data (Table 2) with a spatial resolution of 0.5° × 0.5° to match that of the
lowest resolution dataset (P deposition). Mean annual temperature (MAT), mean annual
precipitation (MAP) and the seasonal temperature amplitude were derived from the global
climate database WorldClim (Fick and Hijmans, 2017). We extracted the Köppen climate
classification to represent different climate zones from the TRY database and filled data gaps



using the {Kgc} R package (Bryant et al., 2017), which provides the Köppen climate
classification for each latitude and longitude. We calculated mean annual evapotranspiration
(ET) and growing season length (GSL) from FLUXCOM (Jung et al., 2011), in which GSL
was based on the seasonal phasing of gross primary productivity (GPP) considering the time
period between 20% and 80% of maximum GPP in an average year for the period 2002-2015.
Total soil P concentrations were derived from Yang et al. 2013; soil clay content and soil pH
were extracted from the Harmonized World Soil Database (HWSD; Wieder et al., 2014). We
used atmospheric N deposition values from CESM-CMIP6 (Hegglin; Kinnison and
Lamarque, 2016) taking the year 2010 as a reference considering that the fields are relatively
smooth, summing the emissions and making the annual mean, and P deposition was extracted
from Brahney et al. (2015) and Chien et al. (2016). All variables used as predictors of global
N and P resorption are described in table 2.

**Table 2.** All possible predictors for nutrient resorption.

|  | Variable name | Unit | Reference |
|---|---|---|---|
| MAT | Mean Annual Temperature | °C | Fick and Hijmans, 2017 |
| MAP | Mean Annual Precipitation | mm | Fick and Hijmans, 2017 |
| AmplT | Temperature amplitude | °C | Fick and Hijmans, 2017 |
| ET | Evapotranspiration | mm | Jung et al., 2011 |
| N_dep2010 | Nitrogen deposition | kgN ha yr | Hegglin; Kinnison and Lamarque, 2016 |
| P_dep | Phosphorus deposition | kgN ha yr | Brahney et al., 2015; Chien et al., 2016 |
| soilP_tot | Total soil P | g P/m$^2$ | Yang et al., 2013 |
| Clay | Top soil clay content | % weight | Wieder et al., 2014 |
| pH | Top soil pH | -log(H+) | Wieder et al., 2014 |
| GSL | Growing season length | days | Jung et al., 2011 |
| SLA | Specific leaf area | mm$^2$ mg$^{-1}$ | Kattge et al., 2020 |
| LLS | Leaf Longevity | month | Kattge et al., 2020 |
| Leaf habit(phenology) | Deciduous/Evergreen | - | Kattge et al., 2020 |
| Leaf Type | Broadleaves/Needles | - | Kattge et al., 2020 |


## 160 2.2 Data derivation

We define nutrient resorption efficiency (NuRE) as the amount of nutrient resorbed during
leaf senescence calculated as:

$$NuRE = \left(1 - \frac{Nu_{senesced}}{Nu_{green}} MLCF\right) \times 100 \tag{1}$$





where $Nu_{green}$ and $Nu_{senesced}$ are nutrient (N or P) concentrations in dry green and senesced leaves (mg g), respectively; MLCF (unitless) is the mass loss correction factor during senescence to account for the loss of leaf mass when senescence occurs. Omitting MLCF overestimates nutrient concentration in senescent leaves and underestimates resorption values (Zhang et al., 2022). Zhang et al. (2022) showed a significant overall improvement when considering MLCF, where both average of N and P resorption increased by ~9%, particularly for cases with low resorption efficiencies. In the present study, not considering the MLCF also underestimates the actual nutrient resorption efficiency when comparing the fraction of resorption of four sub datasets from the final global dataset (Appendix A).

We calculated MLCF as the ratio between the dry mass of senesced and green leaves (van Heerwaarden et al., 2003a), where it was not directly available as percentage leaf mass loss (LML) in the data. We derived average values of MLCF per plant type from nutrient resorption dataset to fill missing values: 0.712 for deciduous, 0.766 for evergreen, 0.69 for conifers, and 0.75 for woody lianas, respectively. To fill in MLCF values for the remaining leaf nutrient and litter data from TRY, we associated these means of MLCF with leaf habit, leaf type and growth form information available on each species. For that, trees with needle evergreen leaves were associated with conifers MLCF; deciduous trees/shrubs with deciduous woody MLCF, and evergreen trees/shrubs with evergreen woody MLCF, respectively. We grouped climbers and lianas with shrubs. In total we extracted data from 131 sites for NRE and 74 for PRE (Fig. 1), with more than one entry per site giving a total of 954 and 454 data points for NRE and PRE species-level, respectively. Temperate biomes were most strongly represented in the dataset (518 entries), followed by tropical (180), boreal (103), polar (102) and dry ecosystems (65).

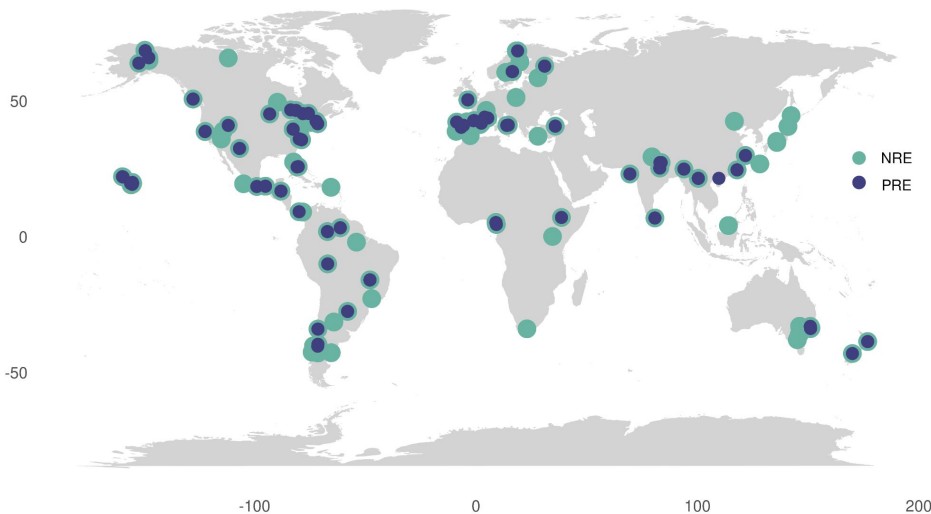

**Figure 1:** Global distribution of data used for nitrogen resorption efficiency (NRE) and phosphorus resorption efficiency (PRE).

## 2.3 Statistical analysis

As the nutrient resorption data did not conform to a normal distribution (Shapiro–Wilk test), we used the nonparametric Kruskal–Wallis one-way ANOVA test of variance to examine differences of NRE and PRE among different climate zones, and Mann-Whitney Wilcoxon test to evaluate differences between leaf habit, leaf type and growth form (deciduous vs evergreen plants, broad-leaves vs needle-leaves, shrubs vs trees), using the {ggstatsplot} R package (Patil, 2021). We applied Pearson correlation and linear regression to analyze the relationship between nutrient resorption and the predictors described in Table 2. For MAP and N deposition, we performed a log transformation prior to conducting the analysis to have the distribution close to the normal. To find the best set of predictors for the variance in NRE and PRE, we used multimodel inference (MMI; Burnham and Anderson, 2002) using the Akaike's information criterion (AIC) and estimated the relative importance of each explanatory variable. Different from setting only a single model based on AIC, multimodel inference accounts for uncertainties in the model performance and in the considered parameters. This approach involves modeling and evaluating all possible combinations of a predetermined set of predictors. The evaluation is typically conducted using a criterion, such



as AIC or Bayesian information criterion (BIC), which favors simpler models and allows for a comprehensive examination of all possible models and their respective performances. By synthesizing the estimated coefficients of predictors across these models, MMI enables inference regarding the overall importance of specific predictors. Before applying MMI, we used generalized linear mixed effect models (GLMER) to fit different models after removing drivers described in Table 2 that showed: (1) high collinearity between them (R ≥ 0.7; Fig. S5); (2) non-significant correlation with NRE (soil P) and PRE (MAP and SLA) (Fig. S5); (3) a threshold of Variance Inflation Factor (VIF) higher than 10 (James et al. 2013). Specifically, temperature amplitude, GSL and ET were not considered due to their high correlation with MAT and MAP and due to high VIF. Based on ecological interactions, we fitted the model considering interactions between climate variables MAT and MAP, as well as between plant characteristics such as leaf structure, leaf habit and leaf type (SLA:LeafPhenology:LeafType). If the ratio between the sample size and the number of parameters considered was higher than 40, we fitted the model using Restricted Maximum Likelihood REML and AICc (corrected for small sample sizes) to avoid bias. We selected the model with lowest AIC and applied it into the 'dredge' function implemented in the multimodal inference package {MuMIn} (Bartoń K, 2023) which generated a full submodel set. A set of best-performing models for NRE and PRE was selected using a cut-off of ΔAIC < 2, and based on these top models, the best model parameters were generated. Using {MuMIn} package, we also calculated the relative importance of each predictor through the sum of the Akaike weights across all models in which the respective parameter was being considered, with a cut-off of 0.8 to distinguish between important and unimportant predictors (Terrer et al., 2016). We performed all statistical analysis using p-value < 0.05 as statistically significant.

## 3. Results

### 3.1 Global patterns of nutrient resorption between different climate zones

The global median of nutrient resorption for nitrogen (NRE) and phosphorus (PRE) is 60.0% ± 12.3% of standard deviation (n=954) and 61.2% ± 13.6% (n=454), respectively. We find differences for both NRE and PRE between the climate zones (Fig. 2). Tropical regions show the lowest resorption for N (NRE: 52.4% ± 12.1%) and tundra ecosystems in polar regions



show the highest (NRE: 69.6% ± 12.8%) (Fig. 2a). PRE in temperate regions shows the
lowest values (57.8% ± 13.6%). PRE increases towards the higher latitude with significant
difference of P resorption from temperate to boreal regions (67.3% ± 13.6%) (Fig. 2b). In
contrast to NRE, the difference of PRE between tropical and other climate zones, as well as
polar regions, is not statistically significant (P > 0.05). NRE in dry regions (61.6% ± 9.7%) is
statistically different from tropical and polar regions, while for PRE, the difference is not
significant between climate zones. However, the sample for this zone is substantially smaller.
Details of minimum, maximum, and median values can be found in Table B1.

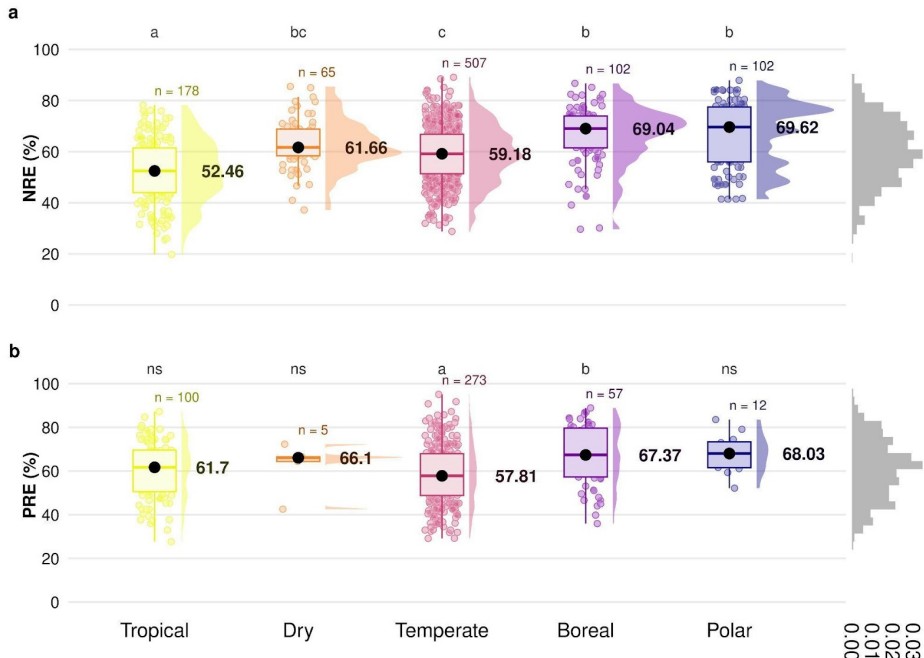

**Figure 2:** Difference in the resorption efficiency of nitrogen (NRE; a) and phosphorus (PRE; b) between climate
zones by Köppen climate classification. Different letters indicate the significant differences in nutrient
resorption between the climate zones, 'ns' means non significant, and 'n' represents the number of observations.

**3.2 Patterns of nutrient resorption between plant functional types**
We explore the variation of nutrient resorption between plant functional groups. Deciduous
woody plants have a significantly higher NRE (65.2% ± 12.4%, n=400) than evergreens
(57.9% ± 11.4%, n=551) (P < 0.001) (Fig. 3a), and shrubs have a significantly higher NRE
(63.1% ± 12.4%, n=230) than trees (59.2% ± 12.1%, n=724) (P < 0.001) (Fig. 3c).



Conversely, there is no significant difference in NRE between broad- (59.8% ± 12.5%,
n=841) and needle-leaved plants (61.8% ± 9.9%, n=103) (P > 0.05) (Fig. 3b). PRE does
neither differ significantly between deciduous (60.0% ± 12.8%, n=220) and evergreen plants
(61.7% ± 14.4%, n=231) (P = 0.4) (Fig. 3d) nor between shrubs (64.4% ± 13.5%, n=59) and
trees (61.1% ± 13.6%, n=395) (P = 0.2) (Fig. 3f). However, PRE differs significantly between
leaf types, with needle-leaved showing higher resorption (72.2% ± 9.2%, n=45) than
broad-leaved plants (59.6% ± 13.5%, n=404) (P < 0.001) (Fig. 3e). Details of minimum,
maximum and median values can be found in Table B2.

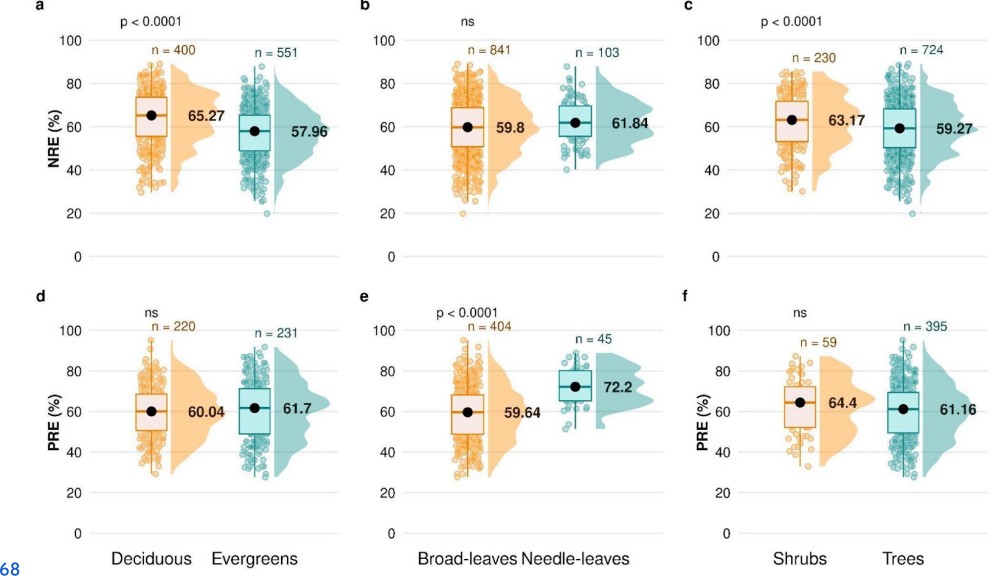

**Figure 3:** Difference in the nitrogen resorption efficiency (NRE) and phosphorus resorption efficiency (PRE)
between plant functional types (PFTs) on a global scale, comparing deciduous versus evergreens (a d),
broadleaved species versus needle leaves (b e), and shrubs versus trees (c f). 'n' represents the number of
observations, and 'p' indicates the significant difference of nutrient resorption between each PFT.
We next explore how climate zones affect NRE and PRE within plant functional groups. NRE
tends to increase from tropical to boreal climates (Fig. 4a) – a pattern seen among deciduous
and evergreen woody plants, among shrubs and trees, and among broadleaved, but not
needle-leaved plants. Also PRE increases from temperate to boreal and polar climates, but
declines from the tropics to temperate climates in evergreens (Fig. 4b). Apart from the overall
tendency, we observe a few statistical deviations from the general pattern that emerged across





all plants pooled: NRE is significantly lower in polar regions compared to boreal forests for
evergreens (NRE: 56.0% ± 13.4%; NRE: 70.5% ± 10.8%) and compared to needle leaved
plants (NRE: 56.0% ± 11.5%; NRE: 51.5% ± 7.3%) (P < 0.001); PRE shows the same pattern
deviation between these regions, but the pattern is not statistically significant (P > 0.05).
Also, we did not observe lower NRE for tropical regions in needle leaved plants because the
only observation of this plant type is in this climate zone. Details of minimum, maximum and
median values can be found in Table B3.

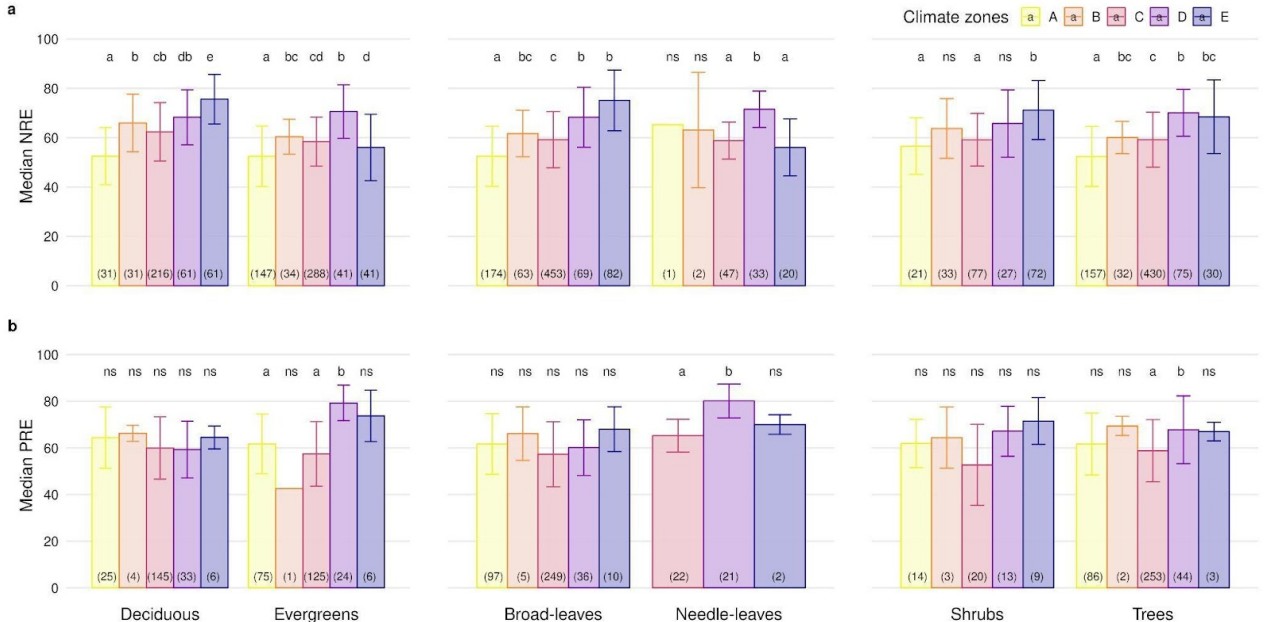

**Figure 4:** Median of nitrogen resorption efficiency (NRE; a) and phosphorus resorption efficiency (PRE; b)
between deciduous versus evergreens, broad- versus needle-leaves and shrubs versus trees in different climate
zones. Error bars are the standard deviations of the medians. Different letters indicate the significant differences
in nutrient resorption between the climate zones. Numbers in parentheses represent the number of observations.
Climate zones (A Tropical; B Dry; C Temperate; D Boreal; E Polar).

## 296 3.3 Main drivers of nutrient resorption

We investigate the main drivers for variation in nutrient resorption, considering biological,
climatic, and soil factors and using data from all PFTs and climate zones pooled. Dredge
model averaging based on a set of best-performing models with corrected AIC (see Methods
2.3) shows that the best model for NRE includes soil clay content, N deposition, MAP and



growth form (Table 3). The best combination of predictors for the PRE model includes N deposition, leaf type, and MAT (Table 3). Sums of Akaike weights indicate that the order of importance of predictors for NRE is N deposition (RI 0.99), MAP (RI 0.99), leaf habit (RI 0.98), followed by soil clay content (RI 0.97), growth form (RI 0.93) and leaf type (RI 0.87) (Fig. 5a); while for PRE, the order is P deposition (RI 0.99), leaf type (RI 0.99), N deposition (RI 0.94) followed by leaf habit (RI 0.89) (Fig. 5b). The criteria to fit the model selecting and/or excluding predictors and interactions for the multimodel inference can be found in Sect. 2.3. Correlations between all variables, as well as linear relationships with the regression slope between nutrient resorption and all possible predictors can be found in Figs. C1 and C2.

**Table 3 |** Summarized results of dredge model averaging for nitrogen resorption efficiency (NRE) and phosphorus resorption efficiency (PRE). Significant codes: 0 '***' 0.001 '**' 0.01 '*' 0.05 '.' 0.1 ' ' 1. SE means standard error.

| NRE | Estimate | SE | Adjusted SE | z value | Pr(>\|z\|) | |
|---|---|---|---|---|---|---|
| (Intercept) | 63.24 | 2.86 | 2.87 | 21.96 | <0.001 | *** |
| Clay content | -0.33 | 0.09 | 0.09 | 3.54 | <0.001 | *** |
| Growth Form | 2.57 | 1.11 | 1.12 | 2.30 | 0.02 | * |
| Leaf habit | 2.02 | 2.32 | 2.33 | 0.86 | 0.38 | |
| Leaf type | 0.66 | 2.51 | 2.52 | 0.26 | 0.79 | |
| MAP | -5.07 | 1.58 | 1.58 | 3.19 | 0.001 | ** |
| N deposition | 0.57 | 0.11 | 0.11 | 5.07 | <0.001 | *** |
| Leaf habit:Leaf type | -0.51 | 2.69 | 2.70 | 0.19 | 0.84 | |
| **PRE** | **Estimate** | **SE** | **Adjusted SE** | **z value** | **Pr(>\|z\|)** | |
| (Intercept) | 78.28 | 9.45 | 9.56 | 8.18 | <0.001 | *** |
| Clay content | -0.44 | 0.24 | 0.24 | 1.81 | 0.06 | . |
| Growth Form | -1.35 | 2.99 | 3.03 | 0.44 | 0.65 | |
| Leaf habit | 2.72 | 1.75 | 1.77 | 1.53 | 0.12 | |
| Leaf type | -10.34 | 4.29 | 4.35 | 2.37 | 0.01 | * |
| MAT | 1.08 | 0.49 | 0.49 | 2.18 | 0.02 | * |
| N deposition | -1.77 | 0.54 | 0.54 | 3.23 | 0.001 | ** |
| P deposition | -97.13 | 65.80 | 66.75 | 1.45 | 0.14 | |



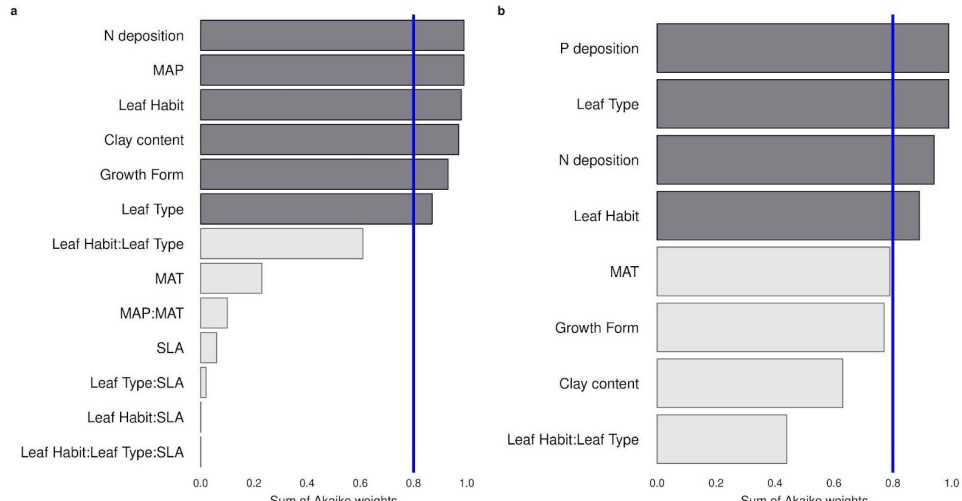

316

**Figure 5:** Importance of the abiotic and biotic predictors on nitrogen resorption efficiency (NRE; a) and phosphorus resorption efficiency (PRE; b). The relative importance (RI) of each predictor is calculated through the sum of the Akaike weights derived from multimodal inference selection, using corrected Akaike's information criteria. The blue line distinguishes between important and unimportant predictors. Mean Annual Precipitation (MAP); Mean Annual Temperature (MAT); SLA (Specific Leaf Area). Colon means interaction between predictors. Leaf habit is represented as 'Leaf Phenology'.

## 4. Discussion

Through an extensive global dataset of leaf nutrient resorption and a multifactorial analysis, we show that leaf habit and type are a strong driver of the spatial variation in nutrient resorption, with thicker, longer-lived leaves having lower resorption efficiencies. Climate, and soil-availability-related factors also emerge as strong drivers, in which we discuss a secondary regulation related to environmental conditions in space and time. Our study covers significantly more woody species observations for nutrient resorption, especially for N, than previous studies (Yuan and Chen, 2009; Yan et al., 2017; Xu et al., 2021). We also account for variations in the mass loss of senescing leaves by deriving the MLCF when leaf mass loss or leaf dry mass were available, and then apply the calculated average MLCF to the missing data, rather than using a single average of MLCF from the literature per PFT (Yan et al., 2017; Xu et al., 2021), which may lead to a more correct estimate of nutrient resorption (see Methods 2.2).



#### 4.1 Nutrient resorption limited by leaf structure

The structural properties of leaves limit the efficiency of resorption along geographic and climatic ranges. We find that the global mean for NRE is significantly higher in deciduous than evergreen plants, and is higher in shrubs than trees (discussed at the end of this section) (Fig. 3a; 3c). This finding is in contrast to previous global studies that found decreasing nutrient resorption with increasing green leaf nutrient content, implying that deciduous species, which generally have higher leaf N content than evergreen species, have higher resorption (Yan et al., 2017; Xu et al., 2021). Nevertheless, our finding is in agreement with Vergutz et al (2013), who reported that deciduous woody species had higher NRE than evergreen woody species and who found no significant differences for PRE.

We find that leaf habit is a strong driver for variation in resorption for both nutrients (Table 3; Fig. 5). Fig. 3a shows that leaf habit is associated with clearly different mean NRE values for evergreen and deciduous species, while the relationship of the average resorption is less clear for PRE (Fig. 3d). This is likely the consequence of a dominance of evergreen species in the tropics in our data set, but we cannot conclude that the lower amount of data for PRE is also a drive of this pattern. The inconsistencies of patterns and significance in P resorption can be related to high biochemical divergence in leaf P fractions compared to N, leading to varied mobilization paths (Estiarte et al., 2023). The breakdown of proteins is the main way N moves around as 75-80% of N is allocated in proteins, while P mobilization involves many different catabolic pathways that lead to wider variety in P dynamics in leaves during leaf development (Estiarte et al., 2023).

We observe no statistical difference between leaf types for NRE (Fig. 3). The higher PRE in needle- than broad-leaves (Fig. 3e) is likely a species effect since almost all needle observations for PRE are plants of the same family, *Pinaceae*. Nevertheless, leaf type is also a strong driver for variance in NRE and PRE (Table 3; Fig. 5). This finding goes together with the view of thicker, longer-lived leaves - such as evergreens and needle-leaves - having lower resorption efficiencies. One possible explanation for this global leaf habit and type pattern is that thicker leaves from evergreens plants, i.e. those with low specific leaf area (SLA), have more N allocated to structural leaf compartments, which means it is harder to break down and resorb nutrients back, leading to less resorption. This is different to deciduous plants, in which leaves are characterized by a higher SLA and a larger N investment into metabolic compounds (Onoda et al., 2017).



The leaf economics spectrum (LES) distinguishes "fast" and "slow" economic strategies found globally and existing independent of climate (Wright et al., 2004). A rapid return on investments, or "fast" economic strategy, is typically associated with deciduous plants and achieved through a combination of traits such as shorter leaf longevity, higher nutrient concentrations, and thinner leaves (high specific leaf area SLA), resulting in higher gas exchange rates per unit mass/area (Reich et al., 1992, 1997; Wright et al., 2004). Conversely, a slow return on investments is associated with the opposite set of traits and typically found in evergreen plants (Reich et al., 1992, 1997; Wright et al., 2004). The low SLA of long-lived leaves is associated with low photosynthetic N-use efficiency, but with nutrient investment spread over a longer period. The low photosynthetic N-use efficiency can be attributed to a higher proportion of C and N being allocated to structural rather than metabolic components of the leaf (Reich et al., 2017), which aligns with the theory on leaf carbon optimization proposed by Kikuzawa (1995) and posits that shorter leaf longevity is associated with higher photosynthetic rates or lower costs of leaf construction.

Here, we found that plants with a conservative nutrient resorption strategy are located at the non-conservative end of the LES, that is, in the "fast" economic strategy. The discussion that revolves around the LES is determined by a combination of trade-offs between investments in structural and metabolic components, as well as trade-offs over time in the expected returns on those investments (Reich et al., 2017). The non-transferable and possibly transferable nutrients depend on where they are located in the cell and their biochemistry (Estiarte et al., 2023). Metabolic fractions are considered to be fully accessible for resorption while structural fractions have been considered non-degradable (Estiarte et al., 2023). Wang et al. (2023) brings the worldwide pattern of high leaf lifespan (LLS) in plants with low SLA as a natural selection response to maximize carbon gain during leaf development, with variations in SLA in deciduous and evergreen species being determined by microclimate conditions. This pattern scales up from the organ level to a broader perspective that encompasses the trade-off between growth and survival at the plant level (Kikuzawa and Lechowicz, 2011). We found higher NRE in shrubs than trees as observed in previous studies (Yuan and Chen, 2009; Yan et al., 2017; Xu et al., 2021), which is also reflected in the identification of plant growth form as one of the main driving factors for NRE in the multimodel inference analysis (Table 3; Fig. 5a). Compared to trees, shrubs typically have smaller leaves and shorter leaf-lifespans. With that they need to be more resourceful with the



nutrients available and prioritize nutrient resorption as a way to optimize nutrient usage for
growth.
Resorption is an internal plant process that aims to maintain the balance of soil-plant
interactions in the acquisition and conservation of nutrients, considering which process is less
costly for the plant. The efficiency in nutrient-use by plants is determined mainly by the
nutrient residence time in the plant, in which they can access through the leaf longevity
maintaining the nutrients or through resorption before leaf abscission (Veneklaas, 2022). Our
results support the concept that nutrient resorption is mainly driven by the share of metabolic
vs total leaf N (P), which co-varies with SLA (proxy for construction costs). Therefore,
higher resorption in deciduous trees may be an important conservation strategy as this
process is less energetically costly than new growth. Brant and Chen (2015) discuss the
dependence of deciduous trees on nutrient resorption efficiency as their investment in green
leaf nutrients is higher to keep fast physiological activity during growing season, or the entire
nutrient economy is compromised. With that, we can argue that leaf longevity may be an
important strategy for evergreen plants to conserve their lower leaf nutrient content, as the
nutrient residence time is higher in evergreens. These plants retain nutrients for as long as
possible, because once the nutrients are transferred to the soil through litterfall, they are
partially lost from the system.

## 4.2 Effects of climate factors

Our global dataset shows that NRE significantly increases from tropical to polar zones (Fig.
2a), while PRE is lowest in temperate zones and significantly increases toward the poles (Fig.
2b). This suggests that the resorption of both nutrients is governed to some extent by a
comparable dependency on climate, possibly related to slowed soil organic matter
decomposition at lower mean annual temperatures, which reduces the net rate of
mineralization and in turn, limits the availability of nutrients for plant uptake from the soil
(Sharma and Kumar 2023). MAT emerges as one of the main drivers for PRE but not for
NRE (Table 3). This result may be the outcome of the overall distribution of deciduous and
evergreen species across climate zones, suggesting that global variations in N and P
resorption along climatic gradients may arise primarily from global patterns in deciduous vs.
evergreen and needle-leaved vs. broadleaved plants. This statement is important in the
context of projecting nutrient cycling under altered climate and indicates limited responses in



resorption to temporal changes in climate at decadal time scales – before the global
distribution of leaf habit and type changes as a result of shifts in species composition.
MAP emerged as an important driver for NRE (Table 3; Fig. 5). One explanation is that low
MAP leads to soil moisture, constraining nutrient mobility and increasing the carbon cost for
plants to take up nutrients (Gill and Penuelas, 2016). Therefore, together with limited N
resorption mobility in leaf tissues discussed above (Estiarte and Penuelas, 2015), soil
moisture constrains N mobilization during the mineralization process (Thamdrup, 2012). Liu
et al. (2016) analyzed the relation between soil N mineralization and temperature sensitivity
on a global scale, and showed largest N mineralization rates at tropical latitudes and a general
poleward decrease. We can observe a similar pattern of NRE with latitude (Fig. C3). Deng et
al. (2018) observed a negative relationship between NRE and mineralisation rate, which
suggests a reciprocal causal relationship where systems emerge exhibiting either
simultaneously low mineralization and high resorption rates. The strong link found here
between NRE and leaf habit and leaf type - traits that are immutable within a given species -
indicates that the variations we observe in resorption might be a possible reflection of species
composition with direct consequence for N cycling. It suggests that a positive feedback
mechanism exists that leads ecosystems to be characterized by high resorption and a slower
soil cycling, or vice versa (Phillips et al. (2013). For example, species adapted to low soil N
are favored in N-limited environments, but they also produce low-N litter that decreases
mineralisation and further favors their competitiveness (Chapin et al., 2011).
In addition, we found a negative correlation between resorption and growing season length
(Figs. C1). Plant strategies in regions with short growing seasons (e,g. high latitudes or
seasonally dry subtropical regions) are focused on nutrient conservation to maximize growth
during the favorable period, despite nutrient availability. In very cold and seasonal
environments, as seen in grassy tundra vegetation, soil nutrients are often not available
concurrently with plant demand (Lacroix et al., 2022), implying that it may be more
advantageous for plants to retain their nutrients. While we did not include growing season
length in the multimodel inference analysis due to its high collinearity with MAT, this aspect
is partially reflected in leaf habit.
When we separate the global patterns for different climate zones in plant functional types
(PFTs), our results show that the major climatic pattern is consistent across the growth forms
and leaf types and leaf habit (Fig. 4), in which NRE and PRE increases towards higher




latitudes and PRE shows a minimum at mid-latitudes. Our findings support that maximum
NRE and PRE may be firstly constrained by leaf properties, with secondary effects from
climate and soil texture (discussed below). Estiarte et al. (2023) suggest that a plant's leaf
biochemistry (biochemical and subcellular fractions of N and P) is the primary factor in
limiting nutrient resorption, followed by secondary regulation related to environmental
conditions in space and time. They present that resorption efficiency declines when soil
nutrient availability rises, as plant uptake becomes less costly in more fertile soil. However,
the expenses linked to aging leaves remain constant (Estiarte et al., 2023).

**476 4.3 Effect of soil nutrient availability**

N and P deposition and clay content emerged as important predictors for both PRE and NRE
(Table 3; Fig. 5). This reflects likely the influence of soil N and P availability for NRE and
PRE. Clay content is an important factor determining the nutrient retention capacity and
cation exchange capacity in soils (Chapin et al., 2011). Chronic N deposition has increased
soil N availability (Galloway et al., 2004) and leaf nutrient content (Chapin et al., 2011) over
the 20$^{th}$ century, and likely affected plant internal recycling and resorption as indicated by our
spatial results. In a fertilization experiment, higher P input had a negative effect on both NRE
and PRE (Yuan & Chen, 2015), suggesting that increased P deposition may reduce the plant
internal recycling and thus resorption. The cycling and accessibility of soil P are influenced
by N deposition (Marklein and Houlton, 2012) through various mechanisms including
changes in plant P use strategies (Dalling et al., 2016; Wu et al., 2020a). Higher N deposition
tends to reduce total soil P content (Sardans et al., 2016) so plants would need to increase
PRE to compensate for the high soil N:P stoichiometry and P limitation. Jonard et al. (2014)
suggests that forest ecosystems are becoming less efficient at recycling P due to excessive N
input and climatic stress. This observation likely contributes to our finding that N and P
deposition emerge as a stronger driver in a negative correlation with PRE (Fig. 5; Figs. C1).
The lack of effect by total soil P on NRE and PRE may result from the fact that this variable
does not represent the actual fraction of P available for plant uptake.
Another soil factor found to be important for nutrient resorption is the clay content (Table 3).
Clay minerals are formed during soil weathering and have high surface area that influences
the soil's water retention capacity, and a negative charge that enables nutrients retention and
exchange with plant roots (Chapin et al., 2011). High-latitude soils that are younger and



experience slow rates of chemical weathering usually have low clay content and therefore, less potential for mineral nutrient storage, which may affect their availability for plant uptake (Chapin et al., 2011). As a result, plants in these environments need to invest more in resorption. Thus, together with MAP and MAT, soil clay content is also closely related to soil nutrient supply on a global scale, which is reflected in its role as driving resorption (Table 3; Fig. 5), as well as in the negative correlation between clay content and nutrient resorption (Figs. C1). In the context of an important effect on nutrient resorption found for leaf properties together with climate, soil texture and soil fertility - previously suggested to be important (Aerts and Chapin, 1999; Yuan and Chen, 2015; Xu et al., 2021) - may indicate that biological and environmental factors are not fully independent, as it is also determined by multiple elements such as litter quality, precipitation, parental materials and soil texture. For example, P availability is geologically and pedologically limited in warm environments, which means mainly determined by soil parent materials (Augusto et al., 2017), and therefore, soil texture becomes an important factor for P limitation in tropical regions. Also, the role of P deposition in relation to plant demand is high for tropical forests (Van Langenhove et al., 2020) but low worldwide (Cleveland et al., 2013). PRE in the tropics did not differ statistically from other climate zones although we observe an increase of PRE from mid to low latitudes (Figs. B1b and C3), which could indicate data limitation for PRE. The combination of plant properties with an underlying soil and climate control as driving factors for resorption variation is also supported by Drenovsky et al. (2010; 2019), who suggested a combination of soil properties, climatic factors, and plant morphology to explain changes in nutrient resorption.

## 4.4 Data uncertainties and implications

Our study contributes to the existing research on nutrient resorption by using a comprehensive approach to derive resorption values from the TRY database. However, we encountered limitations in this derivation due to lack or limited quality of data. The absence of co-located nutrient measurements in leaf and litter led to a shortage of suitable data pairs, mainly for PRE, in which the robustness of the model selection raised concerns about its reliability. While our approach of accounting for the MLCF improved estimates of resorption (Appendix A), we could not estimate the MLCF for all data pairs, and could not fill all gaps using average functional type characteristics due to lacking trait attributes in the TRY



database. These two factors reduced the number of data points available for statistical analysis using multi-model inference. In addition, although recognized the importance of leaf lifespan (LLS), it was not possible to analyze the relationship between resorption and LLS due to the few measurements of this functional trait. Nevertheless, applying the available statistical methods to analyze the drivers behind NRE and PRE, we found consistent patterns for the key gradients of climate, soil and plant functional type, that are informative for other studies despite remaining unexplained variance. In order to improve the depth of resorption investigation, we encourage researchers in field work to perform concurrent measurements of litter nutrient content as well as leaf and litter dry mass.

The statistical analysis of dredge multi-model inference is dependent on the specific factors used in the analysis. We removed highly collinear variables and tested the impact of different combinations of factors. Although such a change in factors affected the exact number of data points used in each multi-model inference, the overall identification of important and less important factors for NRE and PRE was robust, especially for PFTs.

By quantifying these trends that we have found, we can delve deeper into ecosystem models by improving model parametrization and developing a dynamic nutrient resorption concept. Studies that utilize data to infer nutrient cycling frequently simplify resorption making general assumptions (Finzi et al., 2007; Cleveland et al., 2013), or simply representing this process as a fixed value of 50% (Vergutz et al., 2013; Zaehle et al. 2014), which may cause inaccuracies in their findings on nutrient cycling. The flow of recycling nutrients in land surface models is a factor that determines how strong the soil nutrient availability controls plant production. N resorption and N uptake in the FUN model (Fisher et al., 2010), for example, is defined by the relative acquisition cost of the two sources. They discuss that the cost of resorption assumes a constant based on global observations, but it may require a clearer connection to leaf physiology. Here, we provide a start for a statistical model that can connect resorption and plant properties and restrict how much plants could actually resorb nutrients, as well as the dataset to test the predictions of a physiological model. In addition, environmental drivers that have been shown to influence the overall patterns, such as soil texture and climate, could be considered to influence the resorption efficiency after primary leaf physiology limitation. Such information is essential when estimating how it can constrain carbon assimilation in face of global changes (Galloway et al., 2008), and therefore, essential



to predict future plant growth and the capacity of the forest to act as a carbon sink (Thornton et al., 2007; Arora et al., 2022).

## 5. Conclusions

Our analysis of the global plant trait database indicates that variations of NRE and PRE are driven by the combination of plant properties with an additional soil and climate control. Systematic variations of NRE across leaf habit and type indicate that these traits are linked to plant nutrient use and conservation strategies and that leaf structure plays an important role in determining the proportion of nutrients that can be resorbed. Different metrics of soil fertility and soil-related variables were tested and found to have an influence on NRE and PRE together with climatic variables and leaf structure and habit. Clay content, N and P deposition had strong influence with a negative relationship - possibly an expression of its role in nutrient retention - as well as MAP. These trends provide a target to benchmark the simulation of nutrient recycling in global nutrient-enabled models. A focus on considering the links between leaf structure and nutrient resorption efficiency should enable a more realistic consideration of ecological and environmental controls on nutrient cycling and limitation than the current state-of-the-art. The importance of intrinsic plant properties raises important questions about the flexibility of leaf resorption under future changes in climate, $CO_2$ concentrations and atmospheric deposition.

## Acknowledgments

This work was supported by the European Research Council (ERC) under the European Union's Horizon 2020 research and innovation programme (QUINCY; grant no. 647204). BDS was funded by the Swiss National Science Foundation grant PCEFP2_181115. We extend our thanks to our external reviewer Katrin Fleisher, for her helpful comments on the manuscript.

## Author contributions

GS, SC and SZ designed the study. GS performed the analysis. All authors contributed to interpreting the results. GS drafted the manuscripts; all authors contributed to writing and editing the manuscript.



## Data Availability Statement

All data used in this study is publicly available through the TRY database https://www.try-db.org/.

## Conflict of Interests

SZ is a member of the editorial board of Biogeosciences.

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

Availability in Tropical Rain Forests: The Paradigm of Phosphorus Limitation, in: Tropical
Tree Physiology: Adaptations and Responses in a Changing Environment, edited by:
Goldstein, G. and Santiago, L. S., Springer International Publishing, Cham, 261–273, 2016.

Deng, M., Liu, L., Jiang, L., Liu, W., Wang, X., Li, S., Yang, S., and Wang, B.: Ecosystem
scale trade-off in nitrogen acquisition pathways, Nat Ecol Evol, 2, 1724–1734, 2018.

Drenovsky, R. E., James, J. J., and Richards, J. H.: Variation in nutrient resorption by desert
shrubs, J. Arid Environ., 74, 1564–1568, 2010.

Drenovsky, R. E., Pietrasiak, N., and Short, T. H.: Global temporal patterns in plant nutrient
resorption plasticity, Glob. Ecol. Biogeogr., 28, 728–743, 2019.

Du, E., Terrer, C., Pellegrini, A. F. A., Ahlström, A., van Lissa, C. J., Zhao, X., Xia, N., Wu,
X., and Jackson, R. B.: Global patterns of terrestrial nitrogen and phosphorus limitation,
https://doi.org/10.1038/s41561-019-0530-4, 2020.



Elser, J. J., Bracken, M. E. S., Cleland, E. E., Gruner, D. S., Harpole, W. S., Hillebrand, H., Ngai, J. T., Seabloom, E. W., Shurin, J. B., and Smith, J. E.: Global analysis of nitrogen and phosphorus limitation of primary producers in freshwater, marine and terrestrial ecosystems, Ecol. Lett., 10, 1135–1142, 2007.

Estiarte, M., Campioli, M., Mayol, M., and Penuelas, J.: Variability and limits of nitrogen and phosphorus resorption during foliar senescence, Plant Comm, 4, https://doi.org/10.1016/j.xplc.2022.100503, 2023.

Fay, P. A., Prober, S. M., Harpole, W. S., Knops, J. M. H., Bakker, J. D., Borer, E. T., Lind, E. M., MacDougall, A. S., Seabloom, E. W., Wragg, P. D., Adler, P. B., Blumenthal, D. M., Buckley, Y. M., Chu, C., Cleland, E. E., Collins, S. L., Davies, K. F., Du, G., Feng, X., Firn, J., Gruner, D. S., Hagenah, N., Hautier, Y., Heckman, R. W., Jin, V. L., Kirkman, K. P., Klein, J., Ladwig, L. M., Li, Q., McCulley, R. L., Melbourne, B. A., Mitchell, C. E., Moore, J. L., Morgan, J. W., Risch, A. C., Schütz, M., Stevens, C. J., Wedin, D. A., and Yang, L. H.: Grassland productivity limited by multiple nutrients, Nat Plants, 1, 15080, 2015.

Fick, S. E. and Hijmans, R. J.: WorldClim 2: new 1-km spatial resolution climate surfaces for global land areas, Int. J. Climatol., 37, 4302–4315, 2017.

Finzi, A. C., Norby, R. J., Calfapietra, C., Gallet-Budynek, A., Gielen, B., Holmes, W. E., Hoosbeek, M. R., Iversen, C. M., Jackson, R. B., Kubiske, M. E., Ledford, J., Liberloo, M., Oren, R., Polle, A., Pritchard, S., Zak, D. R., Schlesinger, W. H., and Ceulemans, R.: Increases in nitrogen uptake rather than nitrogen-use efficiency support higher rates of temperate forest productivity under elevated $CO_2$, Proc. Natl. Acad. Sci. U. S. A., 104, 14014–14019, 2007.

Fisher, J. B., Sitch, S., Malhi, Y., Fisher, R. A., Huntingford, C., and Tan, S.-Y.: Carbon cost of plant nitrogen acquisition: A mechanistic, globally applicable model of plant nitrogen uptake, retranslocation, and fixation, Global Biogeochem. Cycles, 24, https://doi.org/10.1029/2009gb003621, 2010.

Galloway, J. N., Dentener, F. J., Capone, D. G., Boyer, E. W., Howarth, R. W., Seitzinger, S. P., Asner, G. P., Cleveland, C. C., Green, P. A., Holland, E. A., Karl, D. M., Michaels, A. F., Porter, J. H., Townsend, A. R., and Vöosmarty, C. J.: Nitrogen Cycles: Past, Present, and Future, Biogeochemistry, 70, 153–226, 2004.

Galloway, J. N., Townsend, A. R., Erisman, J. W., Bekunda, M., Cai, Z., Freney, J. R., Martinelli, L. A., Seitzinger, S. P., and Sutton, M. A.: Transformation of the nitrogen cycle: recent trends, questions, and potential solutions, Science, 320, 889–892, 2008.



Güsewell, S.: N : P ratios in terrestrial plants: variation and functional significance, New
Phytol., 164, 243–266, 2004.
Han, W., Tang, L., Chen, Y., and Fang, J.: Relationship between the relative limitation and
resorption efficiency of nitrogen vs phosphorus in woody plants, PLoS One, 8, e83366, 2013.
Hedin, L. O., Brookshire, E. N. J., Menge, D. N. L., and Barron, A. R.: The Nitrogen Paradox
in Tropical Forest Ecosystems, Annu. Rev. Ecol. Evol. Syst., 40, 613–635, 2009.
James, G., Witten, D., Hastie, T., and Tibshirani, R.: An Introduction to Statistical Learning,
Springer US, 15 pp., n.d.
Jonard, M., Fürst, A., Verstraeten, A., Thimonier, A., Timmermann, V., Potočić, N., Waldner,
P., Benham, S., Hansen, K., Merilä, P., Ponette, Q., de la Cruz, A. C., Roskams, P., Nicolas,
M., Croisé, L., Ingerslev, M., Matteucci, G., Decinti, B., Bascietto, M., and Rautio, P.: Tree
mineral nutrition is deteriorating in Europe, Glob. Chang. Biol., 21, 418–430, 2015.
Joswig, J. S., Wirth, C., Schuman, M. C., Kattge, J., Reu, B., Wright, I. J., Sippel, S. D.,
Rüger, N., Richter, R., Schaepman, M. E., van Bodegom, P. M., Cornelissen, J. H. C., Díaz,
S., Hattingh, W. N., Kramer, K., Lens, F., Niinemets, Ü., Reich, P. B., Reichstein, M.,
Römermann, C., Schrodt, F., Anand, M., Bahn, M., Byun, C., Campetella, G., Cerabolini, B.
E. L., Craine, J. M., Gonzalez-Melo, A., Gutiérrez, A. G., He, T., Higuchi, P., Jactel, H.,
Kraft, N. J. B., Minden, V., Onipchenko, V., Peñuelas, J., Pillar, V. D., Sosinski, Ê.,
Soudzilovskaia, N. A., Weiher, E., and Mahecha, M. D.: Climatic and soil factors explain the
two-dimensional spectrum of global plant trait variation, Nat Ecol Evol, 6, 36–50, 2022.
Jung, M., Reichstein, M., Margolis, H. A., Cescatti, A., Richardson, A. D., Arain, M. A.,
Arneth, A., Bernhofer, C., Bonal, D., Chen, J., Gianelle, D., Gobron, N., Kiely, G., Kutsch,
W., Lasslop, G., Law, B. E., Lindroth, A., Merbold, L., Montagnani, L., Moors, E. J., Papale,
D., Sottocornola, M., Vaccari, F., and Williams, C.: Global patterns of land-atmosphere fluxes
of carbon dioxide, latent heat, and sensible heat derived from eddy covariance, satellite, and
meteorological observations, J. Geophys. Res., 116, https://doi.org/10.1029/2010jg001566,
743 2011.
Kattge, J., Díaz, S., Lavorel, S., Prentice, I. C., Leadley, P., Bönisch, G., Garnier, E., Westoby,
M., Reich, P. B., Wright, I. J., Cornelissen, J. H. C., Violle, C., Harrison, S. P., Van
BODEGOM, P. M., Reichstein, M., Enquist, B. J., Soudzilovskaia, N. A., Ackerly, D. D.,
Anand, M., Atkin, O., Bahn, M., Baker, T. R., Baldocchi, D., Bekker, R., Blanco, C. C.,
Blonder, B., Bond, W. J., Bradstock, R., Bunker, D. E., Casanoves, F., Cavender-Bares, J.,
Chambers, J. Q., Chapin, F. S., Iii, Chave, J., Coomes, D., Cornwell, W. K., Craine, J. M.,
Dobrin, B. H., Duarte, L., Durka, W., Elser, J., Esser, G., Estiarte, M., Fagan, W. F., Fang, J.,



Fernández-Méndez, F., Fidelis, A., Finegan, B., Flores, O., Ford, H., Frank, D., Freschet, G. T., Fyllas, N. M., Gallagher, R. V., Green, W. A., Gutierrez, A. G., Hickler, T., Higgins, S. I., Hodgson, J. G., Jalili, A., Jansen, S., Joly, C. A., Kerkhoff, A. J., Kirkup, D., Kitajima, K., Kleyer, M., Klotz, S., Knops, J. M. H., Kramer, K., Kühn, I., Kurokawa, H., Laughlin, D., Lee, T. D., Leishman, M., Lens, F., Lenz, T., Lewis, S. L., Lloyd, J., Llusià, J., Louault, F., Ma, S., Mahecha, M. D., Manning, P., Massad, T., Medlyn, B. E., Messier, J., Moles, A. T., Müller, S. C., Nadrowski, K., Naeem, S., Niinemets, Ü., Nöllert, S., Nüske, A., Ogaya, R., Oleksyn, J., Onipchenko, V. G., Onoda, Y., Ordoñez, J., Overbeck, G., et al.: TRY - a global database of plant traits, Glob. Chang. Biol., 17, 2905–2935, 2011.

Kattge, J., Bönisch, G., Díaz, S., Lavorel, S., Prentice, I. C., Leadley, P., Tautenhahn, S., Werner, G. D. A., Aakala, T., Abedi, M., Acosta, A. T. R., Adamidis, G. C., Adamson, K., Aiba, M., Albert, C. H., Alcántara, J. M., Alcázar C, C., Aleixo, I., Ali, H., Amiaud, B., Ammer, C., Amoroso, M. M., Anand, M., Anderson, C., Anten, N., Antos, J., Apgaua, D. M. G., Ashman, T.-L., Asmara, D. H., Asner, G. P., Aspinwall, M., Atkin, O., Aubin, I., Baastrup-Spohr, L., Bahalkeh, K., Bahn, M., Baker, T., Baker, W. J., Bakker, J. P., Baldocchi, D., Baltzer, J., Banerjee, A., Baranger, A., Barlow, J., Barneche, D. R., Baruch, Z., Bastianelli, D., Battles, J., Bauerle, W., Bauters, M., Bazzato, E., Beckmann, M., Beeckman, H., Beierkuhnlein, C., Bekker, R., Belfry, G., Belluau, M., Beloiu, M., Benavides, R., Benomar, L., Berdugo-Lattke, M. L., Berenguer, E., Bergamin, R., Bergmann, J., Bergmann Carlucci, M., Berner, L., Bernhardt-Römermann, M., Bigler, C., Bjorkman, A. D., Blackman, C., Blanco, C., Blonder, B., Blumenthal, D., Bocanegra-González, K. T., Boeckx, P., Bohlman, S., Böhning-Gaese, K., Boisvert-Marsh, L., Bond, W., Bond-Lamberty, B., Boom, A., Boonman, C. C. F., Bordin, K., Boughton, E. H., Boukili, V., Bowman, D. M. J. S., Bravo, S., Brendel, M. R., Broadley, M. R., Brown, K. A., Bruelheide, H., Brumnich, F., Bruun, H. H., Bruy, D., Buchanan, S. W., Bucher, S. F., Buchmann, N., Buitenwerf, R., Bunker, D. E., et al.: TRY plant trait database - enhanced coverage and open access, Glob. Chang. Biol., 26, 119–188, 2020.

Kikuzawa, K.: Leaf phenology as an optimal strategy for carbon gain in plants, Can. J. Bot., https://doi.org/10.1139/b95-019, 1995.

Kikuzawa, K. and Lechowicz, M. J.: Ecology of leaf longevity, 2011th ed., Springer, Tokyo, Japan, 147 pp., 2011.

Killingbeck, K. T.: Nutrients in senesced leaves: Keys to the search for potential resorption and resorption proficiency, Ecology, 77, 1716–1727, 1996.

Kobe, R. K., Lepczyk, C. A., and Iyer, M.: Resorption efficiency decreases with increasing green leaf nutrients in a global data set, Ecology, 86, 2780–2792, 2005.



Lacroix, F., Zaehle, S., Caldararu, S., Schaller, J., Stimmler, P., Holl, D., Kutzbach, L., and
Goeckede, M.: Decoupling of permafrost thaw and vegetation growth could mean both
ongoing nutrient limitation and an emergent source of N2O in high latitudes, Earth and Space
Science Open Archive, https://doi.org/10.1002/essoar.10510605.1, 2022.
Lam, O. H. Y., Tautenhahn, S., Walther, G., Boenisch, G., Baddam, P., and Kattge, J.: The
"rtry" R package for preprocessing plant trait data,
https://doi.org/10.5194/egusphere-egu22-13251, 2022.
Lang, F., Bauhus, J., Frossard, E., George, E., Kaiser, K., Kaupenjohann, M., Krüger, J.,
Matzner, E., Polle, A., Prietzel, J., Rennenberg, H., and Wellbrock, N.: Phosphorus in forest
ecosystems: New insights from an ecosystem nutrition perspective, J. Plant Nutr. Soil Sci.,
179, 129–135, 2016.
LeBauer, D. S. and Treseder, K. K.: Nitrogen limitation of net primary productivity in
terrestrial ecosystems is globally distributed, Ecology, 89, 371–379, 2008.
Liu, Y., Wang, C., He, N., Wen, X., Gao, Y., Li, S., Niu, S., Butterbach-Bahl, K., Luo, Y., and
Yu, G.: A global synthesis of the rate and temperature sensitivity of soil nitrogen
mineralization: latitudinal patterns and mechanisms, Glob. Chang. Biol., 23, 455–464, 2017.
Luo, Y., Su, B., Currie, W. S., Dukes, J. S., Finzi, A., Hartwig, U., Hungate, B., McMurtrie,
R. E., Oren, R., Parton, W. J., Pataki, D. E., Shaw, R. M., Zak, D. R., and Field, C. B.:
Progressive Nitrogen Limitation of Ecosystem Responses to Rising Atmospheric Carbon
Dioxide, Bioscience, 54, 731–739, 2004.
Marklein, A. R. and Houlton, B. Z.: Nitrogen inputs accelerate phosphorus cycling rates
across a wide variety of terrestrial ecosystems, New Phytol., 193, 696–704, 2012.
Onoda, Y., Wright, I. J., Evans, J. R., Hikosaka, K., Kitajima, K., Niinemets, Ü., Poorter, H.,
Tosens, T., and Westoby, M.: Physiological and structural tradeoffs underlying the leaf
economics spectrum, New Phytol., 214, 1447–1463, 2017.
Patil, I.: Visualizations with statistical details: The "ggstatsplot" approach, J. Open Source
Softw., 6, 3167, 2021.
Phillips, R. P., Brzostek, E., and Midgley, M. G.: The mycorrhizal-associated nutrient
economy: a new framework for predicting carbon-nutrient couplings in temperate forests,
New Phytol., 199, 41–51, 2013.



Reed, S. C., Townsend, A. R., Davidson, E. A., and Cleveland, C. C.: Stoichiometric patterns
in foliar nutrient resorption across multiple scales, New Phytol., 196, 173–180, 2012.
Reich, P. B. and Flores-Moreno, H.: Peeking beneath the hood of the leaf economics
spectrum, New Phytol., 214, 1395–1397, 2017.
Reich, P. B., Walters, M. B., and Ellsworth, D. S.: Leaf Life-Span in Relation to Leaf, Plant,
and Stand Characteristics among Diverse Ecosystems, Ecol. Monogr., 62, 365–392, 1992.
Reich, P. B., Walters, M. B., and Ellsworth, D. S.: From tropics to tundra: global convergence
in plant functioning, Proc. Natl. Acad. Sci. U. S. A., 94, 13730–13734, 1997.
Reich, P. B., Rich, R. L., Lu, X., Wang, Y.-P., and Oleksyn, J.: Biogeographic variation in
evergreen conifer needle longevity and impacts on boreal forest carbon cycle projections,
Proc. Natl. Acad. Sci. U. S. A., 111, 13703–13708, 2014.
Sardans, J., Alonso, R., Janssens, I. A., Carnicer, J., Vereseglou, S., Rillig, M. C.,
Fernández-Martínez, M., Sanders, T. G. M., and Peñuelas, J.: Foliar and soil concentrations
and stoichiometry of nitrogen and phosphorous across E uropean P inus sylvestris forests:
relationships with climate, N deposition and tree growth, Funct. Ecol., 30, 676–689, 2016.
Sharma, P. K. and Kumar, S.: Soil Temperature and Plant Growth, in: Soil Physical
Environment and Plant Growth: Evaluation and Management, edited by: Sharma, P. K. and
Kumar, S., Springer International Publishing, Cham, 175–204, 2023.
Sun, X., Li, D., Lü, X., Fang, Y., Ma, Z., Wang, Z., Chu, C., Li, M., and Chen, H.:
Widespread controls of leaf nutrient resorption by nutrient limitation and stoichiometry,
Funct. Ecol., 37, 1653–1662, 2023.
Tang, L., Han, W., Chen, Y., and Fang, J.: Resorption proficiency and efficiency of leaf
nutrients in woody plants in eastern China, J Plant Ecol, 6, 408–417, 2013.
Terrer, C., Vicca, S., Hungate, B. A., Phillips, R. P., and Prentice, I. C.: Mycorrhizal
association as a primary control of the $CO_2$ fertilization effect, Science, 353, 72–74, 2016.
Thornton, P. E., Lamarque, J.-F., Rosenbloom, N. A., and Mahowald, N. M.: Influence of
carbon-nitrogen cycle coupling on land model response to $CO_2$fertilization and climate
variability, Global Biogeochem. Cycles, 21, https://doi.org/10.1029/2006gb002868, 2007.



Van Heerwaarden, L. M., Toet, S., and Aerts, R.: Current measures of nutrient resorption efficiency lead to a substantial underestimation of real resorption efficiency: facts and solutions, Oikos, 101, 664–669, 2003.

Van Langenhove, L., Verryckt, L. T., Bréchet, L., Courtois, E. A., Stahl, C., Hofhansl, F., Bauters, M., Sardans, J., Boeckx, P., Fransen, E., Peñuelas, J., and Janssens, I. A.: Atmospheric deposition of elements and its relevance for nutrient budgets of tropical forests, Biogeochemistry, 149, 175–193, 2020.

Veneklaas, E. J.: Phosphorus resorption and tissue longevity of roots and leaves – importance for phosphorus use efficiency and ecosystem phosphorus cycles, Plant Soil, 476, 627–637, 2022.

Vergutz, L., Manzoni, S., Porporato, A., Novais, R. F., and Jackson, R. B.: Global resorption efficiencies and concentrations of carbon and nutrients in leaves of terrestrial plants, Ecol. Monogr., 82, 205–220, 2012.

Wang, H., Prentice, I. C., Wright, I. J., Warton, D. I., Qiao, S., Xu, X., Zhou, J., Kikuzawa, K., and Stenseth, N. C.: Leaf economics fundamentals explained by optimality principles, Sci Adv, 9, eadd5667, 2023.

Wickham, H., Averick, M., Bryan, J., Chang, W., McGowan, L., François, R., Grolemund, G., Hayes, A., Henry, L., Hester, J., Kuhn, M., Pedersen, T., Miller, E., Bache, S., Müller, K., Ooms, J., Robinson, D., Seidel, D., Spinu, V., Takahashi, K., Vaughan, D., Wilke, C., Woo, K., and Yutani, H.: Welcome to the tidyverse, J. Open Source Softw., 4, 1686, 2019.

Wieder, W.: Regridded Harmonized World Soil Database v1.2, https://doi.org/10.3334/ORNLDAAC/1247, 2014.

Wright, I. J., Reich, P. B., Westoby, M., Ackerly, D. D., Baruch, Z., Bongers, F., Cavender-Bares, J., Chapin, T., Cornelissen, J. H. C., Diemer, M., Flexas, J., Garnier, E., Groom, P. K., Gulias, J., Hikosaka, K., Lamont, B. B., Lee, T., Lee, W., Lusk, C., Midgley, J. J., Navas, M.-L., Niinemets, U., Oleksyn, J., Osada, N., Poorter, H., Poot, P., Prior, L., Pyankov, V. I., Roumet, C., Thomas, S. C., Tjoelker, M. G., Veneklaas, E. J., and Villar, R.: The worldwide leaf economics spectrum, Nature, 428, 821–827, 2004.

Wu, H., Xiang, W., Ouyang, S., Xiao, W., Li, S., Chen, L., Lei, P., Deng, X., Zeng, Y., Zeng, L., and Peng, C.: Tree growth rate and soil nutrient status determine the shift in nutrient-use strategy of Chinese fir plantations along a chronosequence, For. Ecol. Manage., 460, 117896, 2020.



Xu, M., Zhu, Y., Zhang, S., Feng, Y., Zhang, W., and Han, X.: Global scaling the leaf
nitrogen and phosphorus resorption of woody species: Revisiting some commonly held
views, Sci. Total Environ., 788, 147807, 2021.
Yan, T., Zhu, J., and Yang, K.: Leaf nitrogen and phosphorus resorption of woody species in
response to climatic conditions and soil nutrients: a meta-analysis,
https://doi.org/10.1007/s11676-017-0519-z, 2018.
Yang, X., Post, W. M., Thornton, P. E., and Jain, A.: The distribution of soil phosphorus for
global biogeochemical modeling, Biogeosciences, 10, 2525–2537, 2013.
Yuan, Z. Y. and Chen, H. Y. H.: Global-scale patterns of nutrient resorption associated with
latitude, temperature and precipitation, Glob. Ecol. Biogeogr., 18, 11–18, 2009.
Yuan, Z. Y. and Chen, H. Y. H.: Negative effects of fertilization on plant nutrient resorption,
Ecology, 96, 373–380, 2015.
Yuan, Z.-Y., Li, L.-H., Han, X.-G., Huang, J.-H., Jiang, G.-M., Wan, S.-Q., Zhang, W.-H., and
Chen, Q.-S.: Nitrogen resorption from senescing leaves in 28 plant species in a semi-arid
region of northern China, J. Arid Environ., 63, 191–202, 2005.
Zaehle, S.: Terrestrial nitrogen-carbon cycle interactions at the global scale, Philos. Trans. R.
Soc. Lond. B Biol. Sci., 368, 20130125, 2013.
Zaehle, S., Medlyn, B. E., De Kauwe, M. G., Walker, A. P., Dietze, M. C., Hickler, T., Luo,
Y., Wang, Y.-P., El-Masri, B., Thornton, P., Jain, A., Wang, S., Warlind, D., Weng, E., Parton,
W., Iversen, C. M., Gallet-Budynek, A., McCarthy, H., Finzi, A., Hanson, P. J., Prentice, I.
C., Oren, R., and Norby, R. J.: Evaluation of 11 terrestrial carbon-nitrogen cycle models
against observations from two temperate Free-Air CO2 Enrichment studies, New Phytol.,
202, 803–822, 2014.
Zhang, M., Luo, Y., Meng, Q., and Han, W.: Correction of leaf nutrient resorption efficiency
on the mass basis, J Plant Ecol, 15, 1125–1132, 2022.





**Appendix A - Sensitivity study of the importance of MLCF**


We assembled the global dataset from the gap-filled version of TRY Plant Trait database
(https://www.try-db.org, Kattge et al., 2020, version 5.0) containing field measurements of
paired leaf and litter mass-based tissue N and P concentrations ($N_{mass, leaf}$, $P_{mass, leaf}$, $N_{mass, litter}$,
$P_{mass, litter}$) to derive the fractional nutrient resorption (described in Methods Sect. 2.1).
In order to understand the importance of considering MLCF in the formula to derive reliable
nutrient resorption values, we compared four sub datasets from the final global dataset:
(a) we derived nutrient resorption from nutrient resorption database, in which MLCF was
calculated directly from leaf dry mass or leaf mass loss measurements;
(b) the second dataset we derived nutrient resorption from nutrient resorption database as
well, but we filled the missing values of MLCF using the mean for each plant functional type:
0.712 for deciduous, 0.766 for evergreen, 0.69 for conifers, and 0.75 for woody lianas,
respectively.
(c) the third dataset we derived nutrient resorption using leaf nutrient and litter data from
TRY traits, in which we did not include MLCF in the formula, calculated as:
$$NuRE = \left(1 - \frac{Nu_{senesced}}{Nu_{green}}\right) \times 100 \tag{2}$$
(d) the fourth dataset we derived nutrient resorption using leaf nutrient and litter data from
TRY, but here we filled MLCF with the mean per PFT calculated before, in which we
associated these means with leaf phenology, leaf type and growth form information. For that,
trees with needle evergreen leaves received conifers MLCF, deciduous trees/shrubs received
deciduous woody MLCF, and evergreen trees/shrubs received evergreen woody MLCF,
respectively.
Figure A1 shows nitrogen resorption efficiency (NRE) between different climate zones,
where we can see underestimated values of resorption only when we do not consider MLCF
in the formula (Fig. A1c), with values around or lower 50% of N resorption. We can see more
reliable resorption values around 60% when considering MLCF in the formula (Fig. A1a A1b
A1d). When applying the mean of MLCF for the table deriving NRE from TRY traits (Fig.
A1d), we could reproduce a similar pattern compared to the resorption database imported
from TRY (Fig. A1a). Figure A2 shows the distribution of NRE for each subset described
before, where we can see a clear difference in data distribution only when we do not consider



MLCF in the formula (Fig. A2c). For our final dataset, we then considered together the
dataset (b) and (d), in which are the most reliable data for nutrient resorption as it is providing
more data points for resorption, as well as it is considering MLCF in the formula.

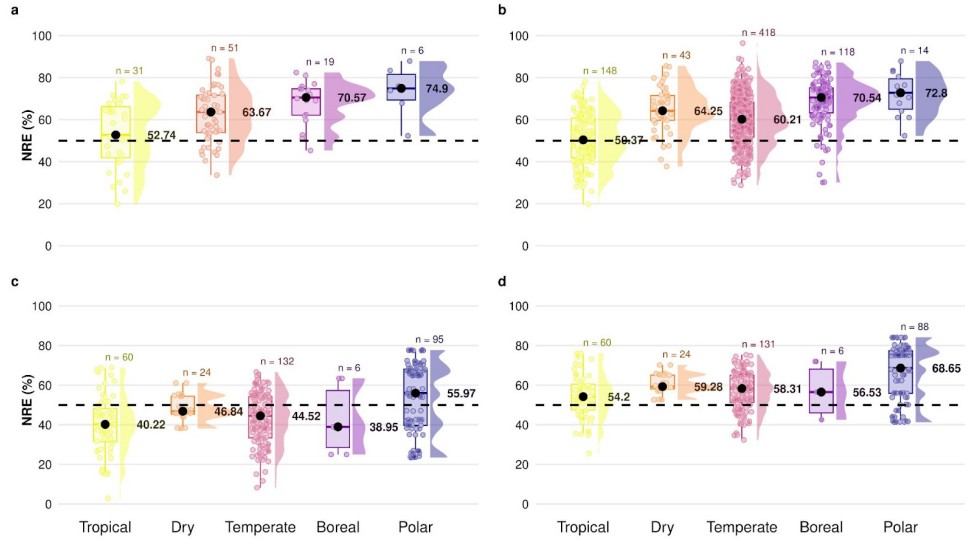



**Figure A1:** Nitrogen resorption efficiency (NRE %) between climate zones by Köppen climate classification.
(a) nutrient resorption values derived directly from nutrient resorption dataset, with MLCF calculated from leaf
dry mass or leaf mass loss measurements; (b) nutrient resorption values derived directly from nutrient resorption
dataset, but with missing MLCF filled by the mean for each plant functional type; (c) nutrient resorption values
derived from TRY traits with no MLCF in the formula; (d) nutrient resorption values derived from TRY traits,
but with missing MLCF filled by the mean for each plant functional type.


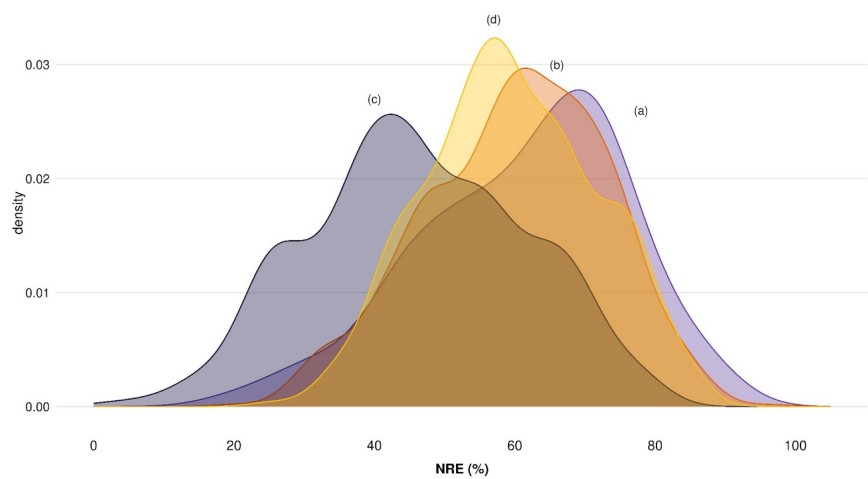




**Figure A2:** Distribution of Nitrogen resorption efficiency (NRE %) for all subsets: (a) nutrient resorption values derived directly from nutrient resorption dataset, with MLCF calculated from leaf dry mass or leaf mass loss measurements; (b) nutrient resorption values derived directly from nutrient resorption dataset, but with missing MLCF filled by the mean for each plant functional type; (c) nutrient resorption values derived from TRY traits with no MLCF in the formula; (d) nutrient resorption values derived from TRY traits, but with missing MLCF filled by the mean for each plant functional type.

# Appendix B - Global patterns of nutrient resorption efficiency for N and P by PFTs and climate zones

**Table B1 |** Summary of Nitrogen resorption efficiency (NRE; %) and Phosphorus resorption efficiency (PRE; %) in different climate zones. For each relationship, the number of observations (N), minimum (Min), maximum (Max), median, and standard deviation (SD) were reported. Letters in Significance show the statistical comparison between each climate zone.

| Resorption (%) | Climate zone | N | Min | Max | Median | SD | Significance |
|---|---|---|---|---|---|---|---|
| NRE | Tropical | 178 | 19.77 | 78.23 | 52.46 | 12.15 | a |
| | Dry | 65 | 37.17 | 85.48 | 61.66 | 9.72 | bc |
| | Temperate | 507 | 28.77 | 89.11 | 59.18 | 11.06 | c |
| | Boreal | 102 | 29.64 | 86.72 | 69.03 | 11.0 | b |
| | Polar | 102 | 41.42 | 87.89 | 69.62 | 12.84 | b |
| PRE | Tropical | 100 | 27.65 | 87.23 | 61.7 | 12.84 | ns |
| | Dry | 5 | 42.55 | 72.31 | 66.09 | 11.47 | ns |
| | Temperate | 273 | 29.14 | 95.11 | 57.80 | 13.65 | a |
| | Boreal | 57 | 35.92 | 88.88 | 67.36 | 13.65 | b |
| | Polar | 12 | 52.16 | 83.58 | 68.02 | 8.84 | ns |

**Table B2 |** Summary of Nitrogen resorption efficiency (NRE; %) and Phosphorus resorption efficiency (PRE; %) in different plant functional types (PFTs). For each relationship, the number of observations (N), minimum (Min), maximum (Max), median, p value and standard deviation (SD) were reported. 'p-value' < 0.05 indicates statistical significance.

| Resorption (%) | PFT | N | Min | Max | Median | p value | SD |
|---|---|---|---|---|---|---|---|
| NRE | Deciduous | 400 | 29.64 | 89.11 | 65.27 | | 12.48 |
| | Evergreens | 551 | 19.77 | 87.89 | 57.96 | **<0.001** | 11.45 |
| | Broad-leaves | 841 | 19.77 | 89.11 | 59.8 | | 12.53 |
| | Needle-leaves | 103 | 40.19 | 87.89 | 61.84 | 0.05 | 9.97 |
| | Shrubs | 230 | 30.13 | 85.48 | 63.17 | | 12.48 |
| | Trees | 724 | 19.77 | 89.11 | 59.27 | **<0.001** | 12.17 |
| PRE | Deciduous | 220 | 29.22 | 95.78 | 60.04 | | 12.86 |
| | Evergreens | 231 | 27.65 | 91.78 | 61.7 | 0.46 | 14.41 |





| | | | | | | | |
|---|---|---|---|---|---|---|---|
| Broad-leaves | 404 | 27.65 | 95.11 | 59.64 | | 13.50 |
| Needle-leaves | 45 | 51.35 | 88.88 | 72.2 | **<0.001** | 9.23 |
| Shrubs | 59 | 32.97 | 87.23 | 64.4 | | 13.50 |
| Trees | 395 | 27.65 | 95.11 | 61.1 | 0.89 | 13.67 |

**Table B3 |** Summary of Nitrogen resorption efficiency (NRE; %) and Phosphorus resorption efficiency (PRE; %) in different plant functional types (PFT) separated in different climate zones. For each relationship, the number of observations (N), minimum (Min), maximum (Max), median, and standard deviation (SD) were reported. Letters in Significance show the statistical comparison between each climate zone.

| | | | NRE | | | | |
|---|---|---|---|---|---|---|---|
| **PFT** | **Climate zones** | **N** | **Min** | **Max** | **Median** | **SD** | **Significance** |
| Deciduous | Tropical | 31 | 31.97 | 71.80 | 52.53 | 11.64 | a |
| | Dry | 31 | 37.17 | 85.48 | 65.95 | 11.68 | b |
| | Temperate | 216 | 31.95 | 89.11 | 62.39 | 11.84 | cb |
| | Boreal | 61 | 29.64 | 86.72 | 68.28 | 11.17 | db |
| | Polar | 61 | 47.15 | 84.16 | 75.60 | 9.99 | e |
| Evergreens | Tropical | 147 | 19.77 | 78.23 | 52.43 | 12.28 | a |
| | Dry | 34 | 40.97 | 79.57 | 60.42 | 7.06 | bc |
| | Temperate | 288 | 28.77 | 81.56 | 58.40 | 9.93 | cd |
| | Boreal | 41 | 30.13 | 82.44 | 70.57 | 10.87 | b |
| | Polar | 41 | 41.42 | 87.89 | 56.03 | 13.44 | d |
| Broad-leaves | Tropical | 174 | 19.77 | 78.23 | 52.46 | 12.15 | a |
| | Dry | 63 | 37.17 | 85.48 | 61.66 | 9.42 | bc |
| | Temperate | 453 | 28.77 | 89.11 | 59.18 | 11.36 | c |
| | Boreal | 69 | 29.64 | 86.72 | 68.28 | 12.13 | b |
| | Polar | 82 | 41.42 | 84.16 | 75.10 | 12.34 | b |
| Needle-leaves | Tropical | 1 | 65.25 | 65.25 | 65.25 | - | ns |
| | Dry | 2 | 46.60 | 79.65 | 63.13 | 23.37 | ns |
| | Temperate | 47 | 40.19 | 81.56 | 58.80 | 7.45 | a |
| | Boreal | 33 | 51.02 | 82.44 | 71.52 | 7.33 | b |
| | Polar | 20 | 46.76 | 87.89 | 56.03 | 11.58 | a |
| Shrubs | Tropical | 21 | 33.81 | 74.33 | 59.60 | 11.45 | a |
| | Dry | 33 | 37.17 | 85.48 | 63.72 | 12.08 | ns |
| | Temperate | 77 | 31.29 | 80.96 | 59.16 | 10.63 | a |
| | Boreal | 27 | 30.13 | 85.15 | 65.77 | 13.66 | ns |
| | Polar | 72 | 41.42 | 84.16 | 71.16 | 11.92 | b |
| Trees | Tropical | 157 | 19.77 | 78.23 | 52.35 | 12.18 | a |
| | Dry | 32 | 47.10 | 76.26 | 60.08 | 6.59 | bc |
| | Temperate | 430 | 28.77 | 89.11 | 59.18 | 11.13 | c |
| | Boreal | 75 | 29.64 | 86.11 | 70.05 | 9.49 | b |
| | Polar | 30 | 46.76 | 87.89 | 68.44 | 14.89 | bc |



| | | | PRE | | | | |
|---|---|---|---|---|---|---|---|
| **PFT** | **Climate zones** | **N** | **Min** | **Max** | **Median** | **SD** | **Significance** |
| Deciduous | Tropical | 25 | 35.92 | 76.26 | 64.40 | 13.14 | ns |
| | Dry | 4 | 64.40 | 72.31 | 66.29 | 3.44 | ns |
| | Temperate | 145 | 29.22 | 95.11 | 59.95 | 13.32 | ns |
| | Boreal | 33 | 35.92 | 84.33 | 59.31 | 12.18 | ns |
| | Polar | 6 | 59.31 | 71.52 | 64.51 | 4.90 | ns |
| Evergreens | Tropical | 75 | 27.65 | 87.23 | 61.70 | 12.81 | a |
| | Dry | 1 | 42.55 | 42.55 | 42.55 | - | ns |
| | Temperate | 125 | 29.14 | 91.78 | 57.44 | 13.85 | a |
| | Boreal | 24 | 61.38 | 88.88 | 79.26 | 7.58 | b |
| | Polar | 6 | 52.16 | 83.58 | 73.73 | 11.03 | ns |
| Broad-leaves | Tropical | 97 | 27.65 | 87.23 | 61.70 | 12.98 | ns |
| | Dry | 5 | 42.55 | 72.31 | 66.10 | 11.47 | ns |
| | Temperate | 249 | 29.14 | 95.11 | 57.28 | 13.93 | ns |
| | Boreal | 36 | 35.92 | 84.33 | 60.14 | 11.92 | ns |
| | Polar | 10 | 52.16 | 83.58 | 68.03 | 9.63 | ns |
| Needle-leaves | Temperate | 22 | 51.35 | 82.62 | 65.25 | 7.06 | a |
| | Boreal | 21 | 61.38 | 88.88 | 80.14 | 7.22 | b |
| | Polar | 2 | 67.02 | 73.00 | 70.01 | 4.22 | ns |
| Shrubs | Tropical | 14 | 47.85 | 79.97 | 61.95 | 10.39 | ns |
| | Dry | 3 | 42.55 | 66.09 | 64.40 | 13.13 | ns |
| | Temperate | 20 | 32.97 | 87.23 | 52.72 | 17.36 | ns |
| | Boreal | 13 | 46.60 | 82.20 | 67.17 | 10.70 | ns |
| | Polar | 9 | 52.16 | 83.58 | 71.52 | 10.0 | ns |
| Trees | Tropical | 86 | 27.65 | 87.23 | 61.70 | 13.24 | ns |
| | Dry | 2 | 66.49 | 72.31 | 69.40 | 4.11 | ns |
| | Temperate | 253 | 29.14 | 95.11 | 58.78 | 13.35 | a |
| | Boreal | 44 | 35.92 | 88.88 | 67.78 | 14.48 | b |
| | Polar | 3 | 61.11 | 68.68 | 67.03 | 3.97 | ns |




## 1035 Appendix C - Linear regressions of nutrient resorption with environmental

## 1036 and biological factors








**Figura C1.** Linear regression of Nitrogen resorption efficiency (NRE; %) and Phosphorus resorption efficiency (PRE; %) with all possible predictor variables. Environmental predictors: Mean Annual Temperature (MAT), Mean Annual Precipitation (MAP), Evapotranspiration (ET), Temperature amplitude (T amplitude), Nitrogen deposition (N deposition), Phosphorus deposition (P deposition), total soil P (soil P)  soil clay fraction (Soil



Clay), soil pH. Biological predictors: Growing Season Length (GSL), Specific Leaf Area (SLA). R: Pearson
correlation; p < 0.05 indicates statistical significance; N: number of observations.

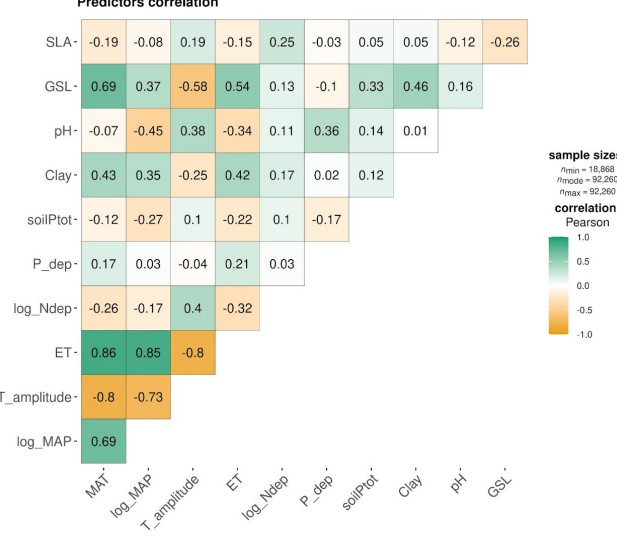


**Figure C2:** Multiple Pearson correlation between all predictors. Mean Annual Temperature (MAT); Mean
Annual Precipitation (MAP); Evapotranspiration (ET); Temperature amplitude (T amplitude); Nitrogen
deposition (N deposition); Phosphorus deposition (P deposition); total soil P (soilPtot);  soil clay fraction (Clay);
soil pH; Growing Season Length (GSL); Specific Leaf Area (SLA).

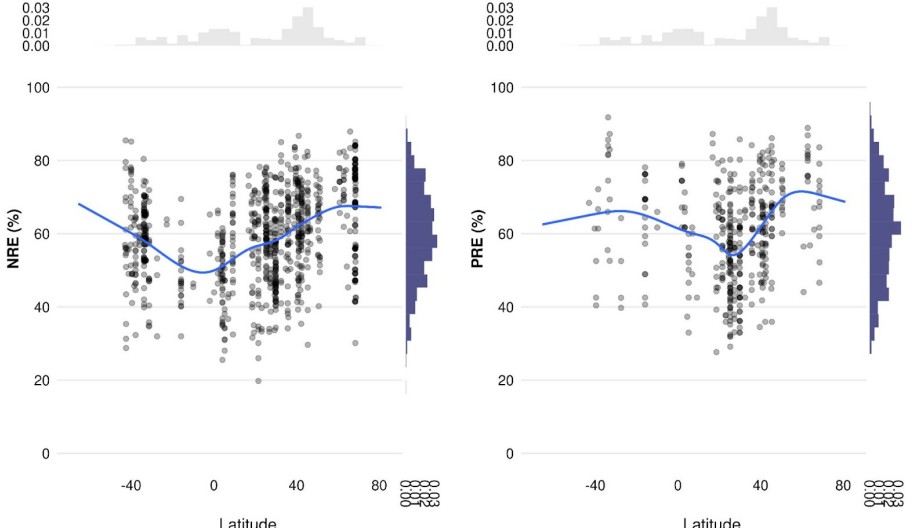


**Figure C3:** Linear regression of nitrogen and phosphorus resorption efficiency (NRE %; PRE %;) with latitude.