# Peer review of "Leaf habit drives leaf nutrient resorption globally alongside nutrient"

_EGUsphere, 2024_

## Author Response (AR1)

**The authors present impressive insights into the major drivers of nutrient resorption at a global scale. This study identifies leaf habit and leaf type as major biotic drivers of nutrient resorption, alongside climate and nutrient-related factors as major abiotic drivers. This finding addresses discrepancies observed in previous studies, offering valuable clarity to the field. As a result, this study deserves broader recognition and could serve as a benchmark for future research. The manuscript is clear, making it easy to read. Therefore, I recommend acceptance of the manuscript, with a few questions that need to be addressed.**

We want to thank the reviewer for the constructive points and for taking the time to carefully read our manuscript. Below, we'll address each of their comments one by one.

**Firstly, considering the authors' findings that different functional groups exhibit varying nutrient resorption efficiencies, I wonder whether these rates are evolutionarily conservative. If species within the same family share similar resorption rates, it could provide valuable insights into functional consequences of climate change and potential shifts in plant communities.**

This point is certainly interesting. We note that we had already included species identity, but not family, as a random factor in the importance analysis model in the previous version, so species identity was incorporated into our final conclusions. We will revise the manuscript to make this clearer.

We looked at the distribution of resorption considering 3 dominant families (shown in the figure below). While the data in our study contains additional families, the limited data available for these additional families makes the magnitude of the spread difficult to interpret. For the three families, the distribution of resorption values within species of the same family is nearly as wide as the distribution for plant functional types (PFTs) when all families are considered together (see Figure 3 of the main text). This finding suggests that even within species of the same family, the observed spread likely reflects a substantial contribution from environmental variability. Estiarte et al., 2023 in their literature review concludes that resorption has substantial interspecific variability being environmentally regulated in space and time. We will add in the data uncertainties and implications section the coordination we tend to see between NRE range of species within the same family and PFTs, which would be interesting for further analysis if more data is available. **(585:588 in the track changes file)**

[Figure]

**The authors selected the best combination of variables based on AIC and BIC values, leading to the exclusion of SLA from the models presented in Table 3. However, from an ecological perspective, understanding the role of SLA while controlling for other environmental factors would be beneficial. Given the authors' observation that "thicker, longer-lived leaves have lower resorption efficiencies," it would be informative to include a model demonstrating the relationship between SLA and nutrient resorption efficiency (NRE or PRE), providing further insight into the mechanisms driving spatial variation in nutrient resorption.**

The significant linear relationship between SLA and NRE (NRE = 55.38 + 0.43 * SLA) is presented in appendix C (note that this relationship is not significant for PRE). Still SLA was not selected for the statistical model according to the AIC and BIC criteria. Instead, plant functional type was selected as an important predictor. In fact, it is possible that this selection is related to the strong relationship between leaf type and SLA. Plant functional type does not appear in the correlation matrix shown in Fig. C1, as it is a categorical variable. The reason for this behavior is the unsurprisingly strong relationship between SLA and PFT in our dataset (figure below), which derives from the leaf economics spectrum (LES) and shows deciduous and broad-leaves with higher SLA than evergreens and needle leaves. We explore the implication of SLA variable on nutrient resorption as part of the results of the categorical variables and using LES theory in the main text where we show e.g. deciduous with higher nitrogen resorption efficiency. We can add this figure below to the SI to support the structural limitations of NRE, but not to the main text as it does not provide any new scientific information. **(339:401 and 1136:1144 in the track changes file)**

[Figure]

**The global analysis presented in this study offers valuable insights. However, there is a need to balance the sample size across climatic zones. It's possible that the dataset over-represents dry regions, which could skew the results. Therefore, I suggest the authors consider a down-sampled dataset to ensure a more balanced representation of different climatic zones. I find it interesting that leaf type significantly influences PRE, and I believe this could represent a facilitative strategy for needle-leaved plants to support cell structures in nutrient-poor habitats. However, upon closer examination of the sample sizes, I am concerned that this result may be biased by the unbalanced distribution of samples across different biomes.**

Thank you for your comment and suggestion. It is true that temperate regions are generally over-represented in global analyses. When applying the model to the dredge function, we already reduce the data availability as we need complete pairs of data when considering all the possible factors that influence NRE(PRE). Replotting Figure 1 of the main text using the data for which the dredge model for NRE was developed (figure below), we observe that the overall pattern and bias for temperate zones remains, but data support becomes challenging. Reducing the amount of data even further will likely not bring more significant and less biased information. We have made an effort to ensure our analysis is as global as possible, and consequently, our statistical dredge model analysis can be influenced by temperate regions bias, which is an inherent limitation we cannot fully mitigate. We will add this information to the discussion of data uncertainties and implications. **(596:599 in the track changes file)**

[Figure]

**Some minor comments:**

**L.185-187: Like what I mentioned above, temperate biomes are over-represented. I wonder if this can influence your analysis result.**

Answer above.

**L.237: you may miss a standard deviation here for NRE.**

We will add the SD for NRE as ± 12.3% in line 237.

**297-310: It would be helpful to see the explanatory powers of the dredge models and separated explanatory powers by biotic and abiotic groups of factors.**

The marginal and conditional values of R2 - used to draw qualitative inferences about the underlying process for the pre-selected models before applying to the dredge function - are 0.23 and 0.98 for the mixed NRE model, and 0.29 and 0.48 for the PRE model. This means both the fixed and random effects explain about 98% of the variance for NRE and 48% of the variance for PRE, in which this variance is attributed to 23% of the fixed effects alone for NRE and 29% for PRE. We will add this information in the SI, but on top of that, the dredge function already selects the best combinations of these models based on the AIC values (< 2) considering the complexity of adding predictors and model performance. I understand your point about the application of models separated by biotic and abiotic groups of factors, however, environmental and biotic factors have strong shared effects in linear mixed models and cannot be separated because they are correlated. **(240:243 and 253:256 in the track changes file)**

**Citation: https://doi.org/10.5194/egusphere-2024-687-RC1**

**Second reviewer's comments**

In this study, Sophia et al. explore the N and P resorption efficiency of woody plants to find global patterns and their main drivers. They conclude that nutrient availability and leaf habit are its main drivers, with a substantial effect of climatic factors.

I found the scope of the study very relevant, and the paper to be comprehensive and beautifully written.

We want to thank the reviewer for the constructive points and for taking the time to carefully read our manuscript. Below, we will address each of their comments one by one.

Nonetheless, I missed further attention to some aspects that I further develop:

1. I appreciate the authors including a discussion about the quality of the data used for the paper. I indeed acknowledge the difficulty of obtaining paired good-quality data on the explored variables. Nonetheless, I believe that there are some further concerns about the data quality or treatment that haven't been reported or discussed:

   1. The percentage of interpolated or gap-filled data included in the analysis should be reported (Line 174-180). Currently, the reader does not have a way to know how much real field data is in there.

      We agree with the reviewer that this information is needed and we will provide this information in the SI together with mass loss correction factor (MLCF) sensitivity. We had 107 observations for NRE and 76 observations for PRE with MLCF derived from real data, considering the mean of MLCF per PFT we were able to have 847 and 378 more observations for NRE and PRE respectively. **(199:201 in the track changes file)**

   2. The time aspect of the data collection should be clarified. Have the green leaf and litter samples been taken at the same time? Did you consider a one-year gap between green leaf and litter? (Since in deciduous trees litter corresponds to the previous season's green leaves).

      It is unfortunately not possible to know the entire temporal aspect of data collection in a database such as TRY, which we agree it's one of its biggest limitations. A significant portion of the data had notes that it was picked from the plant, recently fallen or from litterfall traps cleared every week. For example, for NRE that we have the largest amount of data, 645 of the total 954 observations have this information and are thus collected from the same growing season. We will include this as a limitation in the specific section of the discussion. **(567:571 in the track changes file)**

   3. I assume there is no way to certify that the leaf and litter samples are coming from the same individual and you might have been comparing green leaves and litter from different individuals (even though they are the same species). How big do you think this intraspecific variability is?

      So considering that, I can now answer your third question that approximately 30% of the data could have intraspecific variability. It means that we have leaf and litter measurements for individuals from the same species, but for 30% of the data we cannot confirm that the litter measurement was from the same

growing season and legitimately from the same individual. This is indeed one of the greatest limitations of assessing 'true' nutrient resorption. It is, however, the accepted (and only) method to assess resorption at scale. We will include this limitation in the discussion section. **(571:575 in the track changes file)**

2. **Nutrient reabsorption is also used for nutrient limitation assessment (i.e. Du et al., 2020 cited by authors, or Li et al., 2010…), a concept that authors here refer to as "plant nutrient demand", I believe (line 84-85). The higher the nutrient limitation, the higher the nutrient reabsorption. Therefore, it would be expected that N deposition would be inversely related to NRE, which is what happens with P deposition and PRE. Nonetheless, and according to your reported results, NRE and N deposition are positively correlated, meaning that more N deposition is related to an increase in NRE (Table 3). At the same time, N deposition is considered the most important variable for NRE (Figure 5a). I believe it is a very counterintuitive and interesting result but I couldn't find further discussion about it in section 4.3. Could you please dig further in?**

This is a very important point and we thank the reviewer for pointing out we have not yet discussed this result. It could be either a consequence of a global gradient study in which N deposition is not a good indicator for N limitation as it covaries with climate, although our correlation analysis does not show any strong patterns (Fig. C2). More importantly, it is possible NRE is affected by N deposition via effects on SLA, in which increasing N deposition increases the fraction of non-structurally bound N and therefore increases the fraction of N that can be reabsorbed. This trend, corrected for covariant factors such as leaf type and growth form overlies the hypothesized trend that the fraction of N resorbed given a certain amount of metabolic N increases with nutrient limitation. Our results actually raise an important point in the direct correlation between leaf resorption and nutrient limitation, showing that the relationship is complex and driven by multiple interacting factors and that any direct causality should be used with caution. We will include this explanation. **(524:535 in the track changes file)**

3. **The authors have acknowledged that the data availability to perform this study could be more representative if more data were available. The authors also acknowledged that the results for the needle leave category are mostly based on the Pinaceae family. There is evidence that foliar N and P are strongly influenced by species identity (i.e. Sardans et al., 2021). Perhaps, including the species identity or the phylogenetic distance in your models could improve the reliability of your results. It could be worth exploring.**

We are accounting species identity as a random factor for the mixed effects model to increase the reliability of our results. This detail was not mentioned in the methods, but we will add to the final version. In addition, in the response to reviewer 1, we also show some intra-family variation in resorption. **(240:243 in the track changes file)**

4. **The title does not fully convince me; I don't think it fully represents your conclusions and therefore results. For example, climate is not mentioned even though you conclude to have a significant role (Line 567).**

Thank you for this comment. We'll explore alternative options for the title, such as "Leaf habit together with nutrient availability and climate drives leaf nutrient resorption globally"

**MINOR COMMENTS**

**Line 52: This sentence sounds weird. Maybe changing implies by imply?**

Thank you for this comment. We adjusted the sentence to "The fact that they do not achieve their maximum resorption capacity implies the existence of costs and limitations to resorption." **(line 60 in the track changes file)**

**Line 64: How is "soil fertility" defined? Since it is an important concept for the paper I believe further definition is required.**

Soil fertility was indexed in this paper by N and P deposition and other soil characteristics that globally correlate with nutrient availability, such as total soil P and soil texture. We will write this more clearly in the Methods section of the revised manuscript. **(155:157 in the track changes file)**

**Line 153: Why 2010? Don't you have the year when the data was collected? There are yearly N deposition maps where you could extract the information from. The temporality in Ndep is relevant since it has changed substantially over time, especially in the areas where most of the data is coming from (Ackermann et al., 2019). What do you mean when saying "considering that the fields are relatively smooth"?**

TRY usually provides the year of measurement, but we had no match for nutrient resorption data. The N deposition data is derived from decadal time-slices and derived from initialized CAM runs. Therefore, the information contained in this data set is representative of large-scale features and does not permit investigating local trends. In our view, this would also only be of relevance if the dataset were containing a large fraction of time-dependent resorption estimates, which is not the case. For consistency with P deposition, where we also only have a decadal mean estimate, we chose not to include the trend information.

**Line 343-348: Does this correlate with studies accounting or not for the MLCF?**

Yes, these studies consider MLCF in the formula, however we derive the MLCF when leaf mass loss or leaf dry mass were available, and then apply the calculated average MLCF to the missing data, rather than using a single average of MLCF from the literature per PFT (lines 332 - 335).

**Line 437-438: There might be a mistake on "low MAP leads to soil moisture". It should be "low MAP leads to low soil moisture", right?**

Thank you for this comment, we already corrected the sentence.

**General rambling questions:**

Thank you for your questions, we are glad to extend the discussions on nutrient resorption, however we will not be including these points in the discussion of the paper.

**Can plants make a difference between elements when reabsorbing? For example: If they would be very N limited but P abundant, could they actually "decide" what element to reabsorb?**

This is an interesting physiological question for which our analysis does not and cannot provide any answer. Plants do exhibit selectivity in nutrient resorption, which happens through physiological mechanisms that regulate the internal nutrient recycling based on soil nutrient availability and what the plant's needs considering relative costs evolved with soil nutrient acquisition while the costs of leaf aging remain consistent (Estiarte et al., 2023). So when nitrogen is limited but phosphorus is abundant, plants may prioritize the resorption of N

over P, which is governed by many environmental factors and plant traits and genetics, and processes of regulation within the plant to find the balance for the plant demand. Fertilization experiments for N and P, for example, shows that leaf green Nu and resorption increase if we have an addition of the other element. But I presume that in very limited environments, plants would choose to reabsorb any nutrient to keep the fundamental physiological processes working.

**Do the reabsorption % align with N:P ratios in fresh leaves?**

Our data for N:P and NRE for example indicates a correlation between nutrient resorption and the N:P ratio in fresh leaves. Considering the answer above that the plant aligns internal nutrient recycling with leaf/soil nutrient stoichiometry, this correlation would reflect the discussion on plants' adaptive strategies to optimize nutrient use efficiency in response to the availability of nitrogen and phosphorus and plant's needs. So when the N:P ratio in fresh leaves is low, it indicates that phosphorus is relatively abundant compared to nitrogen, which may result in higher nitrogen resorption efficiency to maintain a balanced nutrient stoichiometry. We will not be including these points in the discussion of the paper.

[Figure]

Ackerman, D., Millet, D. B., & Chen, X. (2019). Global estimates of inorganic nitrogen deposition across four decades. Global Biogeochemical Cycles, 33, 100–107. https://doi.org/10.1029/2018GB005990

XuefengLiX. Li, XingboZhengX. Zheng, ShijieHanS. Han, JunqiangZhengJ. Zheng, and TonghuaLiT. Li. 2010. Effects of nitrogen additions on nitrogen resorption and use efficiencies and foliar litterfall of six tree species in a mixed birch and poplar forest, northeastern China. Canadian Journal of Forest Research. 40(11): 2256-2261.

Sardans, J., Vallicrosa, H., Zuccarini, P. et al. Empirical support for the biogeochemical niche hypothesis in forest trees. Nat Ecol Evol 5, 184–194 (2021). https://doi.org/10.1038/s41559-020-01348-1

Estiarte, M., Campioli, M., Mayol, M., and Penuelas, J.: Variability and limits of nitrogen and phosphorus resorption during foliar senescence, Plant Comm, 4, https://doi.org/10.1016/j.xplc.2022.100503, 2023.

**Helena Vallicrosa**

**Citation: https://doi.org/10.5194/egusphere-2024-687-RC2**

---

## Author Response (AR2)

**Associate editor decision: Publish subject to minor revisions (review by editor)**

**Dear Gabriela Sophia,**

**Your manuscript has been reviewed by the original two reviewers, whom are both happy with the way you have addressed their concerns. There remain just few minor points that should be addressed.**

We want to thank the editor for all the comments and for taking the time to carefully read our manuscript. Below, we will address each comment one by one.

**Since the study is correlational the attribution of causes should be avoided throughout the discussion. Instead of using driver / depends on/ etc., it would be more correct to use 'related to'. There are minor points with the language (e.g., the use of present tense where it would be expected to use simple past, minor concordance issues, typos) that should be corrected.**

Regarding the use of the term 'driver' throughout the paper, this is commonly used in describing such statistical analyses as being synonymous with 'predictor', and is widely understood by the community in its statistical sense and does not imply causality. We have therefore chosen to not change the use of the term.

Regarding the minor points with the language, we reviewed the text and corrected the tense inconsistencies and minor concordance issues we found.

**L167-169 'considering that the fields are relatively smooth'. What does this mean?**

The N deposition data is interpolated to annual from decadal time-slices and derived from initialized CAM runs, therefore, the information contained is representative of large-scale features. (162:165 in the track changes file)

**L374 'have higher resorption'; shouldn't it be 'lower resorption'?**

We corrected the sentence.

**L440-441 'total leaf N (P)'?**

We corrected the sentence.

**L546-550 Revise sentence.**

We made adjustments to the text to enhance its clarity.

**Table 1 - The acronym SLA is repeated four times to identify leaf area per dry mass in leaves with different structure, that could be easily listed under the same acronym. No acronym for 'leaf dry mass' and 'senescent leaf dry mass'? Check the spelling of rachis. Standardise case (minuscule / majuscule).**

**Table 2 – Standardise minuscule / majuscule.**

We adjusted the acronyms and corrected the word in Table 1, and standardized the letter case in both tables.

**Table captions and figure legends should be stand-alone so that they can be understood without recourse to the main text (What? Where? Why?).**

We improved the description of the captions and figure legends by adding more information.

**Revise all acronyms. Sometimes full definitions appear in the text after acronyms had already been defined and other times they are not defined at the first time.**

We corrected the acronyms and definitions throughout the text.

**Citations should be checked throughout the manuscript.**

We checked all the citations and corrected the minor errors.

**Thank you.**

**Kind regards,**

**Erika Buscardo**